# Contextual Rollout Bandits for Reinforcement Learning with Verifiable Rewards

**Xiaodong Lu** [1 2]  **Xiaohan Wang** [2]  **Jiajun Chai** [2]  **Guojun Yin** [2]  **Wei Lin** [2]  **Zhijun Chen** [1]  **Yu Luo** [3]
**Fuzhen Zhuang** [4]  **Yikun Ban** [1 †]  **Deqing Wang** [1 *]

## Abstract

Reinforcement Learning with Verifiable Rewards (RLVR) is an effective paradigm for improving the reasoning capabilities of large language models. However, existing RLVR methods utilize rollouts in an indiscriminate and short-horizon manner: responses of heterogeneous quality within each prompt are treated uniformly, and historical rollouts are discarded after a single use. This leads to noisy supervision, poor sample efficiency, and suboptimal policy updates. We address these issues by formulating rollout scheduling in RLVR as a contextual bandit problem and proposing a unified neural scheduling framework that adaptively selects high-value rollouts throughout training. Each rollout is treated as an arm whose reward is defined by the induced performance gain between consecutive optimization steps. The resulting scheduler supports both noise-aware intra-group selection and adaptive global reuse of historical rollouts within a single principled framework. We provide theoretical justification by deriving sublinear regret bounds and showing that enlarging the rollout buffer improves the achievable performance upper bound. Experiments on six mathematical reasoning benchmarks demonstrate consistent gains in performance and training efficiency across multiple RLVR optimization methods.

## 1. Introduction

From OpenAI O1 (Jaech et al., 2024) to DeepSeek R1 (DeepSeek-AI, 2025), recent advancements (Zou et al.,

2025; Chen et al., 2025b) in large reasoning models have highlighted the crucial role of Reinforcement Learning with Verifiable Rewards (RLVR) in enhancing mathematical reasoning capabilities. By grounding policy optimization in verifiable reward signals, RLVR establishes a scalable self-improvement paradigm that iteratively corrects reasoning trajectories (i.e., rollouts) through trajectory generation and verification-driven learning.

Advanced RLVR methods (Zheng et al., 2025a; Yu et al., 2025a; Shao et al., 2024; Yang et al., 2026) typically employ group-relative policy optimization, which generates multiple responses for a single prompt to derive relative feedback signals. However, these methods often adopt an indiscriminate utilization of responses within a group, incorporating all generated samples into the gradient computation regardless of their individual learning value. In practice, due to the coarse granularity of rule-based rewards, responses within a group exhibit substantial quality heterogeneity, with many providing weak, noisy, or even misleading supervision, such as guessed-correct solutions (Yue et al., 2025; Arcuschin et al., 2025) and trajectories dominated by redundant reasoning steps (Li et al., 2025; Huang et al., 2025a). Failing to filter such low-utility rollouts not only incurs unnecessary computational overhead but can also distort policy updates, motivating the need for fine-grained, noise-aware selection within each rollout group.

Beyond intra-group noise, existing RLVR pipelines also suffer from a second fundamental limitation: *restricted data horizons*. Most methods rely exclusively on the latest batch of rollouts for optimization, discarding past trajectories after a single use. This myopic training paradigm leads to poor sample efficiency and limits attainable policy performance, as many high-value rollouts are never revisited. While recent efforts have explored reusing historical data (Hejna et al., 2024; Zhan et al., 2025), these approaches largely depend on static, hand-crafted heuristics and lack adaptivity to evolving training dynamics. How to systematically and adaptively reuse historical rollouts remains underexplored in RLVR.

To address the above two challenges, we formulate rollout scheduling as a *contextual bandit* problem, where each rollout is viewed as an arm, and the arm reward is derived from the policy performance gain between two consecutive opti-

---

If you have any questions, please contact via *xiaodonglu@buaa.edu.cn* or *yikunb@buaa.edu.cn*. †: Co-advised this work. [1]School of Computer Science and Engineering, Beihang University [2]Meituan [3]Huawei [4]School of Artificial Intelligence, Beihang University. Correspondence to: Deqing Wang <dqwang@buaa.edu.cn>.

mization steps induced by selecting that rollout. Building on this formulation, we propose a neural data scheduler plugin, named CBS, which decomposes naturally into two complementary contributions.

*(i) Intra-group rollout selection.* To address response-quality heterogeneity within groups, CBS introduces a contextual bandit-based scheduler that performs fine-grained selection within each rollout group. We design a compact 10-dimensional arm representation that encodes policy-optimization-related training dynamics (e.g., reward, advantage, entropy, and clipping statistics), and instantiate the scheduler as an MLP that predicts the future utility of each rollout under online feedback. This enables CBS to automatically prioritize informative responses while suppressing noisy or misleading ones during policy optimization.

*(ii) Global Adaptive Rollout Reuse.* To overcome the limited-horizon constraint of existing RLVR methods, CBS further extends the scheduler to a global selection regime over a replay buffer of historical rollouts. By unifying online and offline rollout selection under the same contextual bandit framework, CBS adaptively reuses high-value trajectories across training rounds, substantially improving sample efficiency and enabling long-horizon policy refinement.

We validate the effectiveness of CBS from both theoretical and empirical perspectives. Theoretically, under a simplified yet essential single-rollout selection setting, we establish a formal connection between RLVR data scheduling and contextual bandits, define the per-round regret, and derive a sublinear regret bound for the proposed neural scheduler. We also show that enlarging the replay buffer improves the achievable upper bound of policy performance, providing principled justification for global rollout reuse. Empirically, experiments on six widely used mathematical reasoning benchmarks demonstrate that CBS consistently improves both performance and training efficiency across three representative policy optimization methods. Ablation studies further verify the effectiveness of the proposed components.

Our contributions can be summarized as follows:

- **Unified Contextual-Rollout Bandit for RLVR.** We formulate rollout scheduling in RLVR as a contextual bandit problem and propose a unified neural scheduling framework that supports both (i) noise-aware intra-group rollout selection and (ii) adaptive global reuse of historical rollouts, enabling fine-grained supervision filtering and long-horizon data reuse within a single principled decision-making framework.
- **Theoretical Guarantees.** We establish a formal connection between RLVR data scheduling and contextual bandits, and derive sublinear regret bounds for the proposed scheduler under a simplified yet essential setting, providing theoretical justification for both local and

global rollout selection.
- **Empirical Effectiveness and Efficiency.** Extensive experiments on six mathematical reasoning benchmarks demonstrate that the proposed method consistently improves both performance and training efficiency across multiple RLVR optimization algorithms, with ablation studies validating the contribution of each component.

## 2. Related Work

### 2.1. RLVR for LLM Reasoning

Reinforcement Learning (RL) has emerged as a fundamental paradigm for the post-training of Large Language Models (LLMs), where the model is conceptualized as a policy $\pi_\theta$ that generates tokens as sequential actions to maximize cumulative reward signals. Initial developments in this field primarily focused on Reinforcement Learning from Human Feedback (RLHF) (Ouyang et al., 2022; Dai et al., 2024; Christiano et al., 2017; Rafailov et al., 2023; Azar et al., 2024), which utilizes human preference data to either train reward models (Ouyang et al., 2022) or directly guide policy optimization (Rafailov et al., 2023). More recently, there has been a surge of interest in Reinforcement Learning with Verifiable Rewards (RLVR) for verifiable domains such as mathematics and code generation. These approaches (Zheng et al., 2025a; Yu et al., 2025a; Shao et al., 2024; Yang et al., 2026; 2025b) leverage rule-based rewards—such as deterministic correctness or execution feedback—to incentivize the model's reasoning capabilities and have demonstrated significant potential in improving complex problem-solving tasks. Despite these advancements, contemporary RLVR methods typically utilize generated trajectories in an indiscriminate and local manner, ignoring the heterogeneous learning utility of different rollouts.

### 2.2. Data Selection for RLVR

Data selection is a critical technique for effective LLM post-training, which has been extensively studied in SFT (Zhou et al., 2023; Chen et al., 2023; Lu et al., 2024; Muennighoff et al., 2025; Ivison et al., 2025; Xia et al., 2024) and RLHF (Das et al., 2025; Mahmud et al., 2025; Mehta et al., 2023; Shen et al., 2024; Huang et al., 2025b) scenarios to mitigate data noise and enhance training efficiency. With the rising success of RLVR, recent works have begun to explore data selection mechanisms within this paradigm (Zhang et al., 2025; Zheng et al., 2025b; Hejna et al., 2024; Zhan et al., 2025; Dou et al., 2025; Xu et al., 2025). For instance, GRESO (Zheng et al., 2025b) leverages historical success-failure records of prompts to identify and filter out zero-advantage samples that are unlikely to contribute to policy improvement; while ExGRPO (Zhan et al., 2025) enhances training efficiency by augmenting on-policy data with historical rollouts, specifically prioritizing high-entropy responses

generated under prompts of intermediate difficulty. However, current data selection methods for RLVR predominantly rely on manual and method-specific heuristics that are labor-intensive and lack flexibility, leaving the potential for more adaptive and automated mechanisms largely unexplored.

## 3. Problem Formulation

### 3.1. Preliminary

#### 3.1.1. GROUP RELATIVE POLICY OPTIMIZATION

Group relative policy optimization methods are widely adopted in existing RLVR paradigm. Specifically, given a question-answer pair $(q, a)$, the group relative policy optimization method first generates a group of responses $\{o_i\}_{i=1}^G$ using the current policy. The advantage of each response is then estimated by $A_i = \frac{v_i - \text{mean}(\{v_i\}_{i=1}^G)}{\text{std}(\{v_i\}_{i=1}^G)}$, where $v_i$ is the reward of the $i$-th response. The policy $\pi_\theta$ is optimized by maximizing

$$
\mathcal{J}_\theta = \mathbb{E}_{(q,a)\sim D, \{o_i\}_{i=1}^G \sim \pi_{old}(\cdot|q)} \left[ \frac{1}{G} \sum_{i=1}^G \frac{1}{|o_i|} \sum_{t=1}^{|o_i|} A_i \right.
$$
$$
\left. \Phi(A_i, r_{i,t}(\theta), \epsilon_{\text{low}}, \epsilon_{\text{high}}) \right], \tag{1}
$$

where $\mathcal{D}$ is the training dataset, $\Phi$ is a method-specific clip function, $r_{i,t}(\theta) = \frac{\pi_\theta(o_{i,t}|q,o_{i,<t})}{\pi_{\text{old}}(o_{i,t}|q,o_{i,<t})}$ is the importance sampling weight at the $t$-th token of $o_i$, $\pi_{\text{old}}(\cdot)$ indicates the policy to generate responses $\{o\}_{i=1}^G$, $\epsilon_{\text{low}}$ and $\epsilon_{\text{high}}$ are the hyperparameters controlling the clip lower bound and the clip upper bound respectively. In this paper, we consider three popular policy optimization methods: GRPO (Shao et al., 2024), DAPO (Yu et al., 2025b), and GSPO (Zheng et al., 2025a), and refer to Appendix A for their detailed introduction.

#### 3.1.2. RLVR TRAINING PIPELINE

The RLVR training can be considered as a multi-round policy optimization problem. Assume the standard RLVR training spans $T$ rounds. In each round $t$, a batch of question-answer pairs, $\mathcal{B}_t = \{(q_1^t, a_1^t), \ldots, (q_B^t, a_B^t)\}$, is first randomly sampled from the training dataset $D$, and then a group of responses $\{o_{i,j}^t\}_{j\in[G]}$ are generated by the policy model $\pi_{\theta_{t-1}}$ for each question $q_i$, forming a batch of rollouts $\mathcal{R}_t = \{x_{i,j}^t\}_{i\in[B],j\in[G]}$, where $x_{i,j}^t = (q_i^t, a_i^t, o_{i,j}^t)$ [1] indicates a piece of rollout data. Then the policy parameter

---

is updated by [2]

$$
\theta_t = \theta_{t-1} + \eta_1 \nabla_{\theta_{t-1}} \mathcal{J}_\theta(\mathcal{R}_t), \tag{2}
$$

where $\mathcal{J}_\theta(\mathcal{R}_t)$ indicates the target function in Equation (1), computed using $\mathcal{R}_t$. A visual illustration for this pipeline can be found in Appendix A.2.

### 3.2. Contextual Rollout Bandit

We formulate data scheduling as a contextual bandit problem (Ban et al., 2022). Specifically, we introduce a scheduler that, at each round $t$, selects a high-quality subset $\bar{\mathcal{C}}_t$ with $|\bar{\mathcal{C}}_t| = K$ from a given rollout set $\mathcal{C}_t$. Let $f(\cdot)$ denote a rollout encoding function that maps a subset into a representation. Then, each candidate subset $C' \subseteq \mathcal{C}_t$ of size $K$ can be viewed as an arm with representation $H' = f(C')$. Depending on the candidate arm space and the data source, we study the following two scheduling problems.

> **Definition 3.1 (Intra-Group Scheduling Problem).** We aim to *downsample training rollouts* by retaining only high-quality samples within each group. Formally, for each group of rollouts $\{x_{i,j}^t\}_{j=1}^G \subset \mathcal{R}_t$, we select a top-$p\%$ subset indexed by $\mathcal{S}_i^t \subseteq \{1,\ldots,G\}$ with $|\mathcal{S}_i^t| = \lfloor p\% \cdot G \rfloor$, yielding the selected set $\{x_{i,j}^t\}_{j\in\mathcal{S}_i^t}$. Denoting the selected rollouts as $\bar{\mathcal{C}}_t = \bigcup_{i=1}^B \{x_{i,j}^t\}_{j\in\mathcal{S}_i^t}$, the policy is optimized by training on $\bar{\mathcal{C}}_t$ in Equation (2).

> **Definition 3.2 (Global Scheduling Problem).** We aim to *reuse high-quality historical rollouts*. Specifically, we maintain a FIFO buffer that stores rollouts from the most recent $L$ iterations, $\mathcal{C}_t = \bigcup_{t'=t-L+1}^t \mathcal{R}_{t'}$. At round $t$, we select $K$ high-quality rollouts from $\mathcal{C}_t$ to construct a training set $\bar{\mathcal{C}}_t \subseteq \mathcal{C}_t$, where $|\bar{\mathcal{C}}_t| = K$. We set $K = |\mathcal{R}_t|$ to match the amount of the latest rollouts. The policy is then optimized by training on $\bar{\mathcal{C}}_t$ in Equation (2).

The two scheduling problems will both generate a rollout sequence $\bar{\mathcal{C}}_1, \bar{\mathcal{C}}_2, \cdots, \bar{\mathcal{C}}_T$ and corresponding parameter sequence $\theta_1, \cdots, \theta_T$. Assume we have an unknown reward function $R(\bar{C}_t, \theta_{t-1}, \theta_t)$ that evaluates the performance change from $\pi_{\theta_{t-1}}$ to $\pi_{\theta_t}$, after training on $\bar{\mathcal{C}}_t$. Then the target of the scheduling sequence is to maximize the cumulative reward

$$
R_T = \sum_{t=1}^T R(\bar{C}_t, \theta_{t-1}, \theta_t). \tag{3}
$$

Due to the high storage and computational cost for the LLM policy, it is impractical to accurately evaluate the perfor-

---

[1] We only show partial fields in $x_{i,j}^t$; any quantity with the same index is an attribute of that rollout (i.e., $v_{i,j}^t$ indicates reward).

[2] We adopt the SGD optimizer here for simplicity.

mance change $\theta_{t-1} \to \theta_t$. Therefore, we resort to a computationally efficient surrogate that captures the true performance gain, and define the reward function as the metric gain between two consecutive rollout data $\mathcal{R}_t, \mathcal{R}_{t+1}$.[3]

---

**Definition 3.3 (Performance Gain Reward).** Let $V_t = \text{mean}(\{v_i^t | x_i^t \in \mathcal{R}_t\})$ and $E_t = \text{mean}(\{e_i^t | x_i^t \in \mathcal{R}_t\})$ denote the average reward and entropy for the $t$ step rollout data, with $e_i^t$ being defined as

$$e_i^t = \frac{-1}{|o_i^t|} \sum_{k=1}^{|o_i^t|} \sum_{v \in \mathcal{V}} \pi_{\theta_{t-1}}(v|q_i^t, o_{i,<k}^t) \quad (4)$$
$$\log \pi_{\theta_{t-1}}(v|q_i^t, o_{i,<k}^t),$$

where $\mathcal{V}$ is the vocabulary of the LLM policy. Then the reward function is defined as

$$R(\overline{\mathcal{C}}_t, \theta_{t-1}, \theta_t) = V_{t+1} - V_t - w_e \mathbb{I}[E_t > e_{\min}] * \\ (E_{t+1} - E_t), \quad (5)$$

where $w_e, e_{\min}$ are hyperparameters and $\mathbb{I}[\cdot]$ is an indicator function.

---

Since each mini-batch $\mathcal{B}_t$ is randomly sampled from the training set, $V_t$ provides a stochastic estimate of the expected performance of $\theta_{t-1}$ on the training set $\mathcal{D}$, and thus $V_{t+1} - V_t$ reflects the performance improvement between consecutive updates. The term $w_e \mathbb{I}[E_t > e_{\min}](E_{t+1} - E_t)$ serves as an entropy control term, penalizing entropy increases once the entropy exceeds a predefined threshold $e_{\min}$, preventing training collapse (shown in Section 6.3). It is also worth noting that the deviation between the stochastic estimate and its corresponding expectation is bounded (Lemma B.2).

# 4. Methodology

As shown in Figure 1, we present CBS, a general data selection plugin that can be seamlessly integrated into existing RLVR methods for adaptive rollout selection. CBS employs a neural reward estimator $s_\phi(\cdot)$ that predicts arm-level rewards for rollout selection, and continually adapts its parameters based on online feedback. In the following, we first introduce the overall framework, then describe the detailed configuration, and finally explain how to employ the scheduler to address the two scheduling problems. The overall workflow is summarized in Algorithm 1.

---

[3]From here on, we flatten the two-dimensional indexing into a single dimension, using $x_i^t$ to denote the $i$-th data instance in $\mathcal{R}_t$.

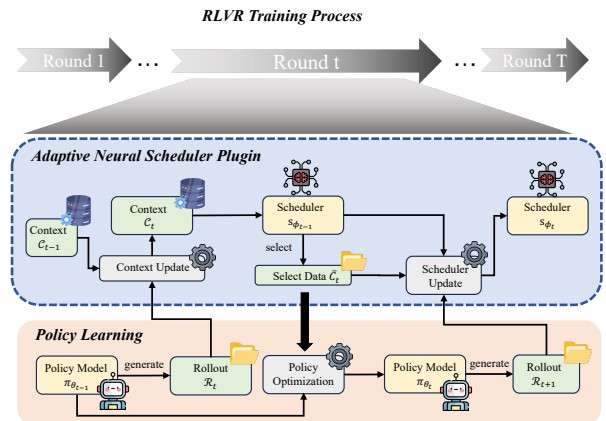

*Figure 1.* Workflow of CBS, introducing a neural scheduler plugin that augments RLVR training by selectively using rollouts. The scheduler also updates based on feedback.

## 4.1. Overall Framework of CBS

**Subset selection via sample scoring.** Scoring each rollout subset $\mathcal{C}' \subset \mathcal{C}_t$ is infeasible due to the combinatorial explosion in the number of arms. We therefore decompose subset-level scoring into per-rollout evaluation. Specifically, let $s_{\phi_{t-1}}$ denote the scheduler at round $t$, for each rollout $x_i^t \in \mathcal{C}_t$, we construct individual representations $h_i^t = f(x_i^t)$ and score it via $s_{\phi_{t-1}}(h_i^t)$. Selecting the optimal size-K rollout subset $\bar{\mathcal{C}}_t$ then reduces to selecting the top-$K$ rollouts. Notably, this decomposition can be justified by a first-order Taylor expansion of the group-level reward, which yields a linear approximation where the overall reward is expressed as the sum of per-sample rewards. Similar linearizations of subset utility are also seen in prior work on data selection for SFT and RLHF, as they enable more efficient subset selection (Xia et al., 2024; Saunshi et al., 2023). Detailed discussion can be found in Appendix C.

**Sample reward.** To provide feedback for the sample scoring, we correspondingly dispatch the group-level reward $R(\bar{\mathcal{C}}_t, \theta_{t-1}, \theta_t)$ to sample-level rewards $R(\bar{x}_i^t, \theta_{t-1}, \theta_t)$ for each $\bar{x}_i^t \in \bar{\mathcal{C}}_t$. Inspired by GRPO-style objectives in Equation (1), where each importance weight is scaled by an advantage term, we adopt a simple advantage-based assignment of:

$$R(\bar{x}_i^t, \theta_{t-1}, \theta_t) = \frac{|\bar{A}_i^t|}{\sum_{\bar{x}_j^t \in \bar{\mathcal{C}}_t} |\bar{A}_j^t|} R(\bar{\mathcal{C}}_t, \theta_{t-1}, \theta_t), \quad (6)$$

where the contribution of each sample is related to its advantage magnitude.

**Scheduler update.** We further update the scheduler parameters online, allowing it to adapt its scheduling policy in response to reward feedback. Specifically, after receiving the sample feedback, we will optimize the scheduler

according to the sample reward, which is

$$\phi_t = \phi_{t-1} - \eta_2 \nabla_{\phi_{t-1}} \frac{1}{K} \sum_{i=1}^{K} \left( s_{\phi_{t-1}}(\bar{h}_i^t) - R(\bar{x}_i^t, \theta_{t-1}, \theta_t) \right)^2, \tag{7}$$

where $\eta_2$ is the learning rate of the scheduler. Since the feedback at round $t$ depends on the rollout in round $t+1$, we perform the scheduler update just after the rollout phase at round $t + 1$ (see Algorithm 1), before scheduling rollouts.

### 4.2. Detailed Configuration

**Arm representation.** A notable amount of prior work has shown that training dynamics associated with roll-out data—such as advantage, entropy, and clip ratio—are strongly correlated with the underlying data value (Zhan et al., 2025; Liu et al., 2025b; Zhu et al., 2025). Inspired by this, we encode each arm $x_i^t$ into a 10-dimensional training-dynamic vector $h_i^t = f(x_i^t) \in \mathbb{R}^{10}$. As shown in Table 1, these features can be divided into four groups of *reward-related signals*, *length-related signals*, *policy perception metrics*, and *data interaction history*. Although each feature is individually simple, their proper combination yields a rich representation of rollouts, making it possible to identify informative patterns, such as high or low entropy under fresh or stale data regimes. Besides, the *policy perception metrics* and *data interaction history* in Table 1 dynamically evolve whenever a rollout is used or a new round begins, enabling a reused rollout to be presented as different representations. Notably, constructing these representations incurs negligible computational overhead (will be shown in Section 6.2): most features are off-the-shelf metrics readily available in standard RL frameworks (e.g., verl), and the relatively complex policy perception metrics can be efficiently obtained during each forward pass without requiring an additional forward computation.

**Scheduler architecture and decision.** We design the scheduler as a multi-layer perceptron (MLP). For scheduling, we select the top-$K$ rollouts based on their scores. Besides, to balance exploration and exploitation, we use epsilon-greedy to select between the highest-scoring and the newest data. Specifically, the $i$-th data $\bar{x}_i^t \in \bar{\mathcal{C}}_t$ is selected by

$$\bar{x}_i^t = \begin{cases} \underset{x \in \mathcal{C}_t \setminus \{\bar{x}_j^t\}_{j=1}^{i-1}}{\arg\max} \ s_{\phi_{t-1}}(h) & \text{w.p. } 1 - \epsilon_t, \\ \underset{x \in \mathcal{C}_t \setminus \{\bar{x}_j^t\}_{j=1}^{i-1}}{\arg\min} \ f_d(x) & \text{w.p. } \epsilon_t. \end{cases} \tag{8}$$

where $h$ indicates the representation of $x$ and $f_d(x) \in \mathbb{R}$ is the *sample age* of $x$ in Table 1. The probability changes as

$$\epsilon_t = \begin{cases} 1.0 & t \le T_w, \\ \max(\epsilon_0 - (t-1) * \Delta_\epsilon, \epsilon_{\min}) & t > T_w, \end{cases} \tag{9}$$

*Table 1.* The 10-dimensional training-dynamics representation is defined for each rollout $x_i^t \in \mathcal{C}_t$.

| Description | Mathematical Formulation |
|---|---|
| **Reward-related signals** *(Intra-group relative merit)* | |
| Reward | $v_i^t$ |
| Advantage | $A_i^t$ |
| Mean group reward | $\text{mean}\left(\{v_j^t \mid x_j^t \text{ is in the same group as } x_i^t\}\right)$ |
| Std of group reward | $\text{std}\left(\{v_j^t \mid x_j^t \text{ is in the same group as } x_i^t\}\right)$ |
| **Length-related signals** *(Response volume and integrity)* | |
| Normalized length [1] | $\lvert o_i^t \rvert / L_{\max}$ |
| Truncation flag | $\mathbb{I}[o_i^t \text{ is truncated}]$ |
| **Policy perception metrics** *(Model state projected onto data)* | |
| Entropy [2] | $-\frac{1}{\lvert o_i^t \rvert} \sum_{j=1}^{\lvert o_i^t \rvert} \sum_{v \in \mathcal{V}} \pi_{\theta_{t-}}(v \mid q_i^t, o_{i,<j}^t) \cdot \log \pi_{\theta_{t-}}(v \mid q_i^t, o_{i,<j}^t)$ |
| Clip ratio | $\frac{1}{\lvert o_i^t \rvert} \sum_{j=1}^{\lvert o_i^t \rvert} \mathbb{I}[r_{i,j}^t(\theta_{t-}) \text{ is clipped}]$ |
| **Data interaction history** *(Usage and freshness)* | |
| Usage count [3] | $\sum_{k=1}^{t-1} \mathbb{I}[x_i^t \in \bar{\mathcal{C}}_k]$ |
| Sample age [4] | $t - \max\{k \mid k < t, o_i^t \in \mathcal{R}_k\}$ |

[1] $L_{\max}$ denotes the maximum response length.
[2] $t^-$ denotes the latest round that $x_i^t$ is used for policy optimization, or the round at which it was generated if it has never been used. The same definition applies to the clip ratio.
[3] The number of times $x_i^t$ is used for policy optimization.
[4] The number of rounds elapsed since the sample was generated.

where $T_w, \Delta_\epsilon, \epsilon_{\min}, \epsilon_0$ are hyperparameters controlling the warmup step, probability decay weight, minimum explore probability, and initial explore probability, respectively.

**Reward augmentation.** To improve the training stability of the scheduler, we normalize the group reward by Exponential Moving Average (EMA). Specifically, we first replace the raw reward gain in (5) with an EMA one and squeeze it within $[0, 1]$ by Sigmoid, which is

$$\hat{R}(\bar{\mathcal{C}}_t, \theta_{t-1}, \theta_t) = \left[ \text{Sigmoid}\left( \frac{(V_{t+1} - V_t) - \mu_t}{\sqrt{\sigma_t}} \right) - w_e \mathbb{I}[E_t > e_{\min}](E_{t+1} - E_t) \right], \tag{10}$$
$$\mu_t = (1 - \alpha)\mu_{t-1} + \alpha(V_{t+1} - V_t),$$
$$\sigma_t = (1 - \alpha)\sigma_{t-1} + \alpha\left[(V_{t+1} - V_t) - \mu_t\right]^2$$

where $\alpha$ is a hyperparameter to control the degree of smoothness. Additionally, we find that the denominator in (6) yields very small sample weights when many rollouts are selected, leading to unstable scheduler training. We therefore remove the denominator and redefine the sample reward as

$$\hat{R}(\bar{x}_i^t, \theta_{t-1}, \theta_t) = \lvert A_i^t \rvert * \hat{R}(\bar{\mathcal{C}}_t, \theta_{t-1}, \theta_t) \tag{11}$$

Finally, we summarize the overall workflow in Algorithm 1.

### 4.3. Employing the Scheduler for the Two Scheduling Problems

**Intra-group scheduling.** For intra-group scheduling, we set the context set to the latest batch of rollouts (i.e., $\mathcal{C}_t = \mathcal{R}_t$) and organize rollouts by group. In each scheduling round, the scheduler respectively selects the top-$p\%$ rollouts within each group and aggregates them to form the selected set $\bar{\mathcal{C}}_t$.

**Global scheduling problem.** For the global scheduling problem, we employ a FIFO replay buffer to store rollouts from the latest $L$ steps and perform global selection to choose $K$ rollouts to form the set $\bar{\mathcal{C}}_t$. Moreover, since the *policy perception metrics* and *data interaction history* in Table 1 evolve across training rounds—where *sample age* increases over time, while other features change only when a rollout is selected for policy optimization—we update these features at the end of each round (see Algorithm 1).

## 5. Theoretical Analysis

In this section, we formally describe the connection between the data scheduling and contextual bandit problem to provide intuitive motivation for CBS. For clarity, we consider the single-sample selection setting, in which we select one rollout per round, and the sample reward coincides with the subset reward. We refer to Appendix C for the analysis of batch selection. For reward function, according to Lemma B.2, the raw scheduler reward satisfies $\mathcal{R}(\bar{\mathcal{C}}_t, \theta_{t-1}, \theta_t) \in [-A, A]$, where $A = 1 + w_e \log N$ and $N = |\mathcal{V}|$ is the vocabulary size. Therefore, in the analysis, we use the normalized reward

$$\bar{R}(\bar{\mathcal{C}}_t, \theta_{t-1}, \theta_t) = \frac{R(\bar{\mathcal{C}}_t, \theta_{t-1}, \theta_t) + A}{2A}.$$

Then $\bar{R}(\bar{\mathcal{C}}_t, \theta_{t-1}, \theta_t) \in [0, 1]$, which satisfies the bounded label condition required by the neural regression lemmas. Notably, this affine normalization only scales each regret gap by the constant factor $1/(2A)$, and thus does not change the order of the regret bound. All proofs for this section can be found in Appendix B.

Under the above setting, Algorithm 1 naturally reduces to a standard contextual bandit problem. Let $M = |\mathcal{C}_t|$ denote the context size. At each round $t$, a set of arms $\mathcal{C}_t = \{x_1^t, \ldots, x_M^t\}$ and their associated representations $H_t = \{h_1^t, \ldots, h_M^t\}$ are presented to the scheduler. An arm $a_t \in [M]$ is selected, and a reward $r_t$ is observed for the chosen arm. Let $V_t^s = \bar{R}(\bar{\mathcal{C}}_t, \theta_{t-1}, \theta_t)$ be the random variable denoting the scheduler reward at round $t$. Rewriting the policy optimization and the expected rewards as functions below,

$$\begin{aligned} \mathcal{A}(x, \theta_{t-1}) &= \theta_{t-1} + \eta \nabla_{\theta_{t-1}} J_\theta(x), \\ g(\theta_t) &= \mathbb{E}_{\mathcal{B}_{t+1} \sim \mathcal{D}, \mathcal{R}_{t+1} \sim \pi_{\theta_t}(\cdot | \mathcal{B}_t)} \left[ V_t^s \right], \end{aligned} \quad (12)$$

the $r_t$ can be considered as a stochastic estimate of function $g(\mathcal{A}(x_{a_t}^t, \theta_{t-1}))$. Therefore, we can define the optimal arm as $a_t^* = \arg\max_{x \in \mathcal{C}_t} g(\mathcal{A}(x, \theta_{t-1}))$. The cumulative regret is then given as

$$R_T = \sum_{t=1}^{T} \left[ g(\mathcal{A}(x_{a_t^*}^t, \theta_{t-1})) - g(\mathcal{A}(x_{a_t}^t, \theta_{t-1})) \right]. \quad (13)$$

In the following, we interchangeably use $g(\mathcal{A}(x_a^t, \theta_{t-1}))$ and $r_a^t$ for notation simplicity.

Our theoretical results build on the Neural Tangent Kernel (NTK) framework, which provides a tractable characterization of training dynamics in the overparameterized regime. We instantiate the selector network $s_\phi(\cdot)$ as a fully connected network with depth $L \geq 2$ and width $m$ (details in Appendix B.1). We slightly abuse the notations and rewrite the arm representations $\{h_i^t\}_{i \in [M], t \in [T]}$ and rewards $\{r_i^t\}_{t \in [T], i \in [M]}$ into $\{h_i\}_{i \in [TM]}$ and $\{r_i\}_{i \in [TM]}$ respectively. Let $H \in \mathbb{R}^{TM \times TM}$ denotes the NTK matrix of $\{h_i\}_{i \in [TM]}$ (defined in Appendix B.1). We adopt the following two standard assumptions for contextual bandit analysis (Zhou et al., 2020; Zhang et al., 2021; Dai et al., 2023; Jia et al., 2022; Ban et al., 2024).

**Assumption 5.1.** For any $i \in [K]$, $t \in [T], \theta \in \mathbb{R}^d$, $\|h_i^t\|_2 = 1$ and $g(\theta) \in [0, 1]$.

**Assumption 5.2.** There exists $\lambda_0 > 0$ that $H \succeq \lambda_0 I$.

The assumptions are mild: the first is satisfied when the RLVR reward is bounded in $[0, 1]$ and the arm representations are $\ell_2$-normalized before being fed into $s_\phi$, while the second holds as long as there are no identical representations among $\{h_i\}_{i \in [TM]}$.

We first investigate how the buffer size of global scheduling impacts the final policy performance. Holding all other hyperparameters fixed, we define $\varphi(M) = \max_\phi \mathbb{E}\left[\max_{t \in [T]} g(\theta_t)\right]$ as the optimal expected performance achievable during the training process, by setting buffer size to $M$ and selecting the scheduler policy $s_\phi$. The expectation is taken with respect to the randomness in the training procedure (i.e., rollout phase). We then state the following theorem.

> **Theorem 5.3.** *Let $B$ denote the batch size and $G$ denote the group size. There exist constants $C_1, C_2 > 0$ such that, if $m \geq C_1$ and $L \geq C_2$, then for any $i, j \in [T]$ with $i \geq j$, it holds that*
>
> $$\varphi(iBG) \geq \varphi(jBG).$$

Theorem 5.3 emphasizes the benefits of reusing historical rollouts, demonstrating that selectively leveraging rollouts from multiple prior rounds—not just the most recent

round—can enhance the performance upper bound of the policy model. Next, we present the regret bound for CBS.

---

**Theorem 5.4.** *Let $r = [r_1, \cdots, r_{TM}] \in \mathbb{R}^{TM}$ be the stacked reward vector. If Assumption 5.1 and Assumption 5.2 hold, for any $\delta \in (0, 1)$, there exists a choice of scheduler hyperparameters $(m, \eta_2)$ and a decay schedule for the exploration probability $\epsilon_t$ such that, with probability at least $1 - \delta$ over scheduler initialization,*

$$\mathbb{E}[R_T] \leq \mathbb{E}\left[\tilde{O}\left(\sqrt{T}\left(\tilde{d} + S\sqrt{\tilde{d}}\right) + \sqrt{T}\right)\right],$$

*where $S = \sqrt{r^T H^{-1} r}$ and $\tilde{d} = \frac{\log \det(I + H)}{\log(1 + TM)}$.*

---

Theorem 5.4 provides a more fine-grained characterization of CBS. First, it shows that the regret increases sublinearly with the time horizon $T$, achieving a data-dependent $\tilde{O}(\sqrt{T})$ regret bound. Second, it reveals that the regret is intrinsically related to the effective dimension $\tilde{d}$ of the neural tangent kernel (NTK) and the geometric structure of the reward captured by $S$, where simpler feature structures (small $\tilde{d}$) and better-aligned rewards (small $S$) will lead to a smaller regret. Notably, in favorable structured regimes, $\tilde{d}$ can remain controlled. For example, (Zhou et al., 2020) (Remark 4.4) and (Zhang et al., 2021) (Remark 3.3) show that when the NTK matrix of arm representations has an underlying low-dimensional structure, $\tilde{d}$ can remain bounded.

# 6. Experimental Results

## 6.1. Experimental Settings

**Models and datasets.** We conduct experiments using Qwen3 Base models (Yang et al., 2025a) of various sizes, namely Qwen3-1.7B-Base, Qwen3-4B-Base, and Qwen3-8B-Base. Following (Yang et al., 2025b), the training set is constructed by combining the DAPO dataset (Yu et al., 2025a) and a subset of the Math dataset (Hendrycks et al., 2021) containing level 3-5 problems, yielding a total of 21723 mathematical problems. For evaluation, we adopt a suite of benchmark datasets, including AIME24, AIME25, AMC23, MATH500 (Hendrycks et al., 2021), Minerva (Lewkowycz et al., 2022), and Olympiad (He et al., 2024).

**Implementation details.** We implement CBS based on verl (Sheng et al., 2025). Maximum response length is set to 4096 for all experiments. Following (Zhan et al., 2025; Yang et al., 2025b), we report Avg@4 for datasets with larger problems (i.e., MATH500, Minerva, Olympiad) and Avg@32 for datasets with smaller problems (i.e., AIME24, AIME25 and AMC23). More details regarding the experimental setup are provided in the Appendix D. The code is available at https://github.com/lxd99/CBS_public.

## 6.2. Main Results

**Intra-group scheduling improves both performance and computational efficiency.** As shown in Table 2, integrating CBS* into representative policy optimization methods consistently yields substantial performance gains across different model sizes, demonstrating the effectiveness and robustness of our adaptive selection strategy. In particular, CBS* achieves average relative improvements of 5.0%, 5.8%, and 2.0% for GRPO, DAPO, and GSPO, respectively. Moreover, Figure 2(b) shows that CBS* reduces the average policy optimization time by 50.4% across the three methods, as fewer training trajectories are required per actor update. These results indicate that prioritizing a subset of high-value rollouts provides a more effective and efficient paradigm for RLVR training compared to using the full rollout set.

**Global scheduling further improves training efficiency.** As shown in Table 2, CBS also consistently enhances different policy optimization methods, yielding relative improvements of 4.0%, 8.5%, and 1.7% for GRPO, DAPO, and GSPO, respectively. Both CBS and CBS* continue to improve performance even when a rule-based data filter is applied (e.g., DAPO with dynamic sampling), suggesting that heuristic selection rules are insufficient to capture the complex and dynamically evolving utility of rollout data during training. Furthermore, Figure 2 (a) shows that CBS achieves higher training rewards than both CBS* and the original policy optimization baseline, indicating that appropriately reusing historical rollouts can further improve training efficiency. Importantly, the computational overhead of the scheduler is negligible: as shown in Table 3, buffer-related operations (e.g., scheduler updates and arm representation construction) account for less than 4% of total training time and decrease further as model size increases. Due to space limitations, more training dynamics curves are provided in Appendix E.1.

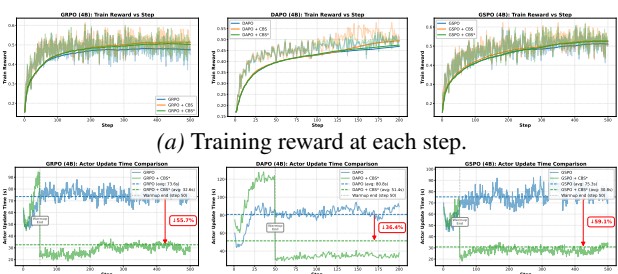

*(a)* Training reward at each step.

*(b)* Policy optimization at each step, where the horizontal dashed line denotes the mean optimization time.

*Figure 2.* Train dynamics of different RLVR methods.

## 6.3. Ablation Study

To validate the effectiveness of the proposed submodules, we conduct an ablation study by comparing CBS with the

*Table 2.* Performance of different RLVR methods when equipped with CBS, where CBS indicates the global scheduler and the CBS* indicates the intra-group scheduler. Relative improvement (Rel. Impr.) is computed w.r.t. the corresponding policy optimization method, based on the *Average* score.

| Method | AIME24 | AIME25 | AMC23 | MATH500 | Minerva | Olympiad | Average | Rel. Impr. |
|---|---|---|---|---|---|---|---|---|
| *Qwen3-1.7B-Base* | | | | | | | | |
| Base Model | 1.77 | 0.73 | 15.31 | 29.10 | 9.83 | 11.09 | 11.31 | – |
| GRPO | 9.58 | 4.38 | 42.97 | 60.95 | 26.29 | 29.30 | 28.91 | – |
| GRPO + CBS | 8.12 | 3.12 | 45.78 | 63.90 | 28.95 | 31.16 | 30.17 | +4.4% |
| GRPO + CBS* | 11.98 | 7.81 | 45.47 | 64.80 | 26.10 | 30.30 | **31.08** | +7.5% |
| DAPO | 6.35 | 4.69 | 37.97 | 58.70 | 20.59 | 27.23 | 25.92 | – |
| DAPO + CBS | 8.96 | 4.90 | 46.48 | 63.65 | 26.01 | 29.97 | **30.00** | +15.7% |
| DAPO + CBS* | 10.52 | 6.04 | 45.16 | 62.45 | 24.82 | 31.01 | 30.00 | +15.7% |
| GSPO | 10.73 | 5.52 | 43.52 | 65.95 | 27.94 | 31.53 | 30.87 | – |
| GSPO + CBS | 11.35 | 5.83 | 46.72 | 65.70 | 28.22 | 33.09 | 31.82 | +3.1% |
| GSPO + CBS* | 12.50 | 7.08 | 43.36 | 67.50 | 30.79 | 33.23 | **32.41** | +5.0% |
| *Qwen3-4B-Base* | | | | | | | | |
| Base Model | 5.10 | 3.65 | 21.33 | 31.90 | 10.85 | 17.54 | 15.06 | – |
| GRPO | 21.04 | 15.21 | 61.56 | 73.70 | 33.55 | 42.32 | 41.23 | – |
| GRPO + CBS | 20.31 | 18.44 | 63.05 | 74.45 | 34.47 | 43.73 | **42.41** | +2.9% |
| GRPO + CBS* | 20.52 | 16.56 | 62.19 | 75.55 | 34.93 | 43.43 | 42.20 | +2.4% |
| DAPO | 16.88 | 17.40 | 62.97 | 75.05 | 31.71 | 43.95 | 41.33 | – |
| DAPO + CBS | 20.62 | 20.94 | 62.89 | 75.85 | 36.58 | 46.70 | **43.93** | +6.3% |
| DAPO + CBS* | 18.96 | 17.81 | 63.83 | 76.30 | 34.83 | 44.88 | 42.77 | +3.5% |
| GSPO | 18.44 | 15.21 | 67.34 | 77.55 | 41.18 | 44.92 | 44.11 | – |
| GSPO + CBS | 21.46 | 18.33 | 67.03 | 76.05 | 36.49 | 44.18 | 43.92 | -0.4% |
| GSPO + CBS* | 21.25 | 19.17 | 66.02 | 77.85 | 34.83 | 46.55 | **44.28** | +0.4% |
| *Qwen3-8B-Base* | | | | | | | | |
| Base Model | 8.33 | 6.56 | 35.78 | 53.25 | 19.67 | 26.22 | 24.97 | – |
| GRPO | 20.52 | 17.29 | 67.34 | 76.15 | 37.50 | 47.14 | 44.32 | – |
| GRPO + CBS | 23.65 | 20.83 | 69.45 | 77.45 | 38.79 | 48.59 | 46.46 | +4.8% |
| GRPO + CBS* | 26.46 | 19.48 | 69.92 | 77.45 | 38.88 | 47.59 | **46.63** | +5.2% |
| DAPO | 22.29 | 19.27 | 73.20 | 77.65 | 37.78 | 48.29 | 46.41 | – |
| DAPO + CBS | 23.54 | 21.88 | 75.55 | 78.25 | 39.71 | 49.41 | **48.06** | +3.6% |
| DAPO + CBS* | 24.38 | 19.58 | 68.52 | 76.85 | 37.50 | 46.96 | 45.63 | -1.7% |
| GSPO | 27.92 | 19.38 | 73.20 | 79.20 | 40.53 | 48.37 | 48.10 | – |
| GSPO + CBS | 23.02 | 20.73 | 73.83 | 83.10 | 45.13 | 49.93 | **49.29** | +2.5% |
| GSPO + CBS* | 25.10 | 20.52 | 72.34 | 79.65 | 42.00 | 51.08 | 48.45 | +0.7% |

*Table 3.* Buffer-related runtime overhead (% of total time)

| Scheduler | Method | 1.7B | 4B | 8B |
|---|---|---|---|---|
| CBS | GRPO | 3.83 | 1.45 | 1.15 |
| | DAPO | 2.88 | 1.37 | 1.00 |
| | GSPO | 1.97 | 1.79 | 1.56 |
| CBS* | GRPO | 1.70 | 0.92 | 0.84 |
| | DAPO | 1.95 | 0.83 | 0.69 |
| | GSPO | 1.98 | 0.90 | 0.79 |

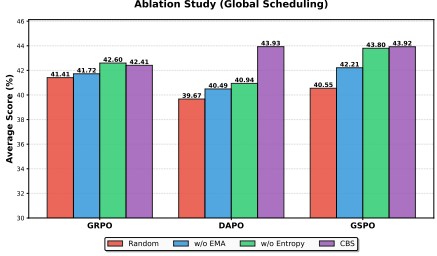

*Figure 3.* Ablation study results on Qwen3-4B-Base.

following variants: (1) replacing the neural scheduler with random selection, denoted as *Random*; (2) replacing the EMA reward gain in Equation (10) with the raw reward gain (i.e., $V_{t+1} - V_t$), denoted as *w/o EMA*; and (3) removing the entropy penalty in Equation (5), denoted as *w/o Entropy*.

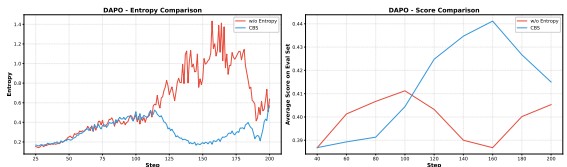

*Figure 4.* Comparison of Entropy and average score on the evaluation set for CBS and w/o Entropy.

As shown in Figure 3, CBS generally achieves superior performance compared to other variants, demonstrating the effectiveness of the proposed submodules. Notably, *Random* degrades the performance of all policy optimization methods, suggesting that data selection is non-trivial, as naive random sampling may introduce unstable or conflicting training signals. In Figure 4, we further compare the entropy dynamics of *w/o Entropy* and CBS. The results show that, without the entropy penalty, the entropy of *w/o Entropy* keeps increasing and the performance deteriorates accordingly, highlighting the necessity of the entropy penalty term. More ablation study results can be found in Appendix E.2.

**Other experimental results.** Due to space limitations, we put other experimental results in the Appendix, including hyperparameter sensitivity analysis (Appendix E.3) and scheduler pattern analysis (Appendix E.4).

## 7. Conclusion

In this paper, we present CBS, a contextual bandit-based scheduler that supports adaptive selection of high-value rollouts for both intra-group and global rollout scheduling problems. By encoding each rollout as a 10-dimensional training dynamics vector and using an MLP-based scheduler that learns adaptively from online policy feedback, CBS effectively prioritizes trajectories with high optimization potential, thereby significantly improving both final performance and training efficiency across three popular RLVR methods. Currently, CBS relies on a lightweight 10-dimensional representation of training dynamics for computational efficiency. Future work could explore integrating deep semantic representations from the LLM policy to uncover more nuanced selection patterns, further unlocking the potential of adaptive rollout scheduling.

## Impact Statement

This paper presents an adaptive rollout scheduler whose goal is to advance the field of Reinforcement Learning with Verifiable Rewards. There are many potential societal consequences of our work, none of which we feel must be specifically highlighted here.

## Acknowledgements

This research was supported by NSFC (No. 62276015), NSFC (No. 62506024) and GW2025-09.

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

# A. Introduction of Group Relative Policy Optimization Algorithms

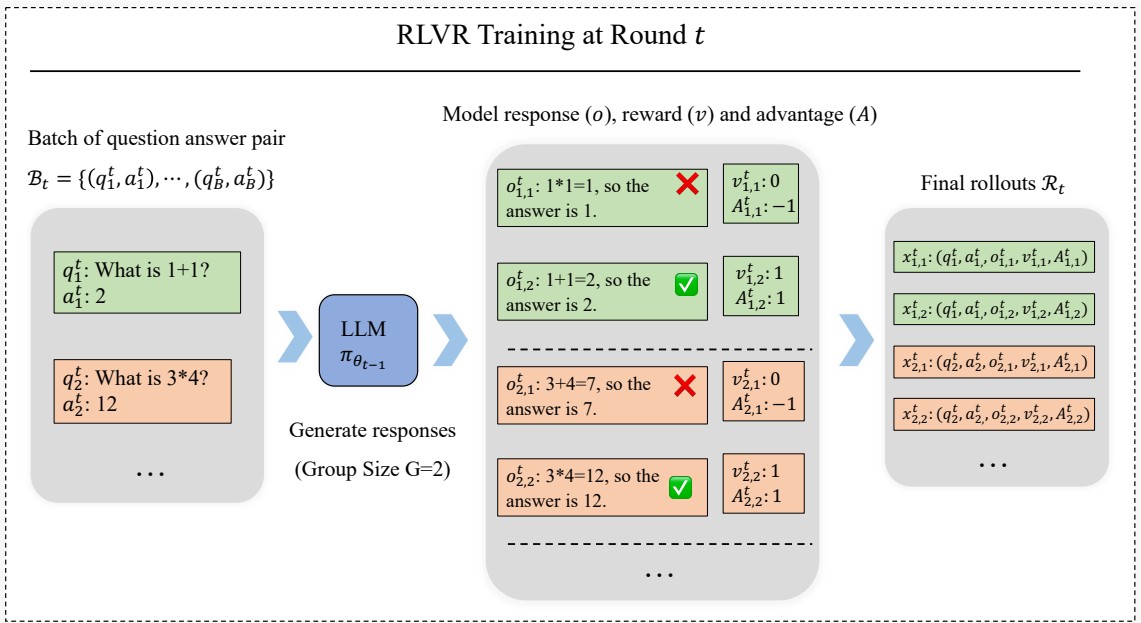

*Figure 5.* Pipeline of RLVR training. At round $t$, a batch of prompt-answer pairs $\mathcal{B}_t = \{(q_i^t, a_i^t)\}_{i \in [B]}$ is first sampled. The current policy $\pi_{\theta_{t-1}}$ then generates a group of responses $\{o_{i,j}^t\}_{j \in [G]}$ for each question $q_i^t$. The corresponding rewards $\{v_{i,j}^t\}_{j \in [G]}$ and advantages $\{A_{i,j}^t\}_{j \in [G]}$ are computed according to responses $\{o_{i,j}^t\}_{j \in [G]}$ and answers $a_i^t$, and the resulting rollout instances together form $\mathcal{R}_t = \{x_{i,j}^t\}_{i \in [B], j \in [G]}$.

## A.1. Target of Different Policy Optimization Methods

Reinforcement Learning is a technique that has been widely applied across many domains (Chen et al., 2024; 2025a; 2026a;b; Yuan et al., 2025). Recently, RLVR has become a critical method for improving the reasoning ability of LLMs (Lin et al., 2026a;c;b; Wang et al., 2026; Li et al., 2026b;a; Zhang et al., 2026; Liu et al., 2026). In this section, we introduce three representative policy optimization methods.

**GRPO (Group Relative Policy Optimization)** is a popular method for policy optimization in language model post-training. It optimizes the policy $\pi_\theta$ by maximizing

$$\mathcal{J}_\theta = \mathbb{E}_{(q,a) \sim D, \{o_i\}_{i=1}^G \sim \pi_{old}(\cdot|q)} \left[ \frac{1}{G} \sum_{i=1}^G \frac{1}{|o_i|} \sum_{t=1}^{|o_i|} \min(A_i r_{i,t}(\theta), A_i \text{clip}(r_{i,t}(\theta), 1 - \epsilon_{\text{low}}, 1 + \epsilon_{\text{high}})) \right], \tag{14}$$

where the $\text{clip}(r, p_{\text{low}}, p_{\text{high}}) = \max\big(p_{\text{low}}, \min(r, p_{\text{high}})\big)$ is a hard clip function. The GRPO can be considered as a variant of Equation (1) by setting $\Phi$ to

$$\Phi(A, r_{i,t}(\theta), \epsilon_{\text{low}}, \epsilon_{\text{high}}) = \min(A_i r_{i,t}(\theta), A_i \text{clip}(r_{i,t}(\theta), 1 - \epsilon_{\text{low}}, 1 + \epsilon_{\text{high}})). \tag{15}$$

**Dynamic Sampling Policy Optimization (DAPO)** introduces four key improvements over GRPO: 1) *Dynamic Sampling*, which filters out zero-advantage responses; 2) *Clip Higher*, which increases the clip upper bound $\epsilon_{\text{high}}$; 3) *Overlong Filter*, which shapes the reward by penalizing overly long responses; and 4) *Token-Level Loss*, which normalizes the loss by the total token length across the group of responses. By combining these components, the policy $\pi_\theta$ is optimized by maximizing

$$\mathcal{J}_\theta = \mathbb{E}_{(q,a) \sim \mathcal{D}, \{o_i\}_{i=1}^G \sim \pi_{\text{old}}(\cdot|q)} \left[ \frac{1}{\sum_{i=1}^G |o_i|} \sum_{i=1}^G \sum_{t=1}^{|o_i|} \min\left( A_i r_{i,t}(\theta), \text{clip}(r_{i,t}(\theta), 1 - \epsilon_{\text{low}}, 1 + \epsilon_{\text{high}}) A_i \right) \right] \tag{16}$$

$$\text{s.t.} \quad 0 < |\{o_i | \text{is\_equivalent}(a, o_i)\}| < G$$

**Algorithm 1** Policy Training with Data Scheduler

1: **Input:** policy $\pi_{\theta_0}$, scheduler $s_{\phi_0}$
2: **for** $t = 1, 2, \ldots, T$ **do**
3:     ▽ Rollout Construction
4:     Randomly select a batch $\mathcal{B}_t$ from training data $\mathcal{D}$
5:     Generate rollout $\mathcal{R}_t$ by current policy $\pi_{\theta_{t-1}}$ and construct corresponding representations in Table 1
6:     Construct $\mathcal{C}_t$ using $\mathcal{R}_t$ and $\mathcal{C}_{t-1}$
7:     ▽ Scheduler Training
8:     Construct sample reward $\hat{R}(x, \theta_{t-2}, \theta_{t-1})$ for $x \in \bar{\mathcal{C}}_{t-1}$ according to Equation (10) and Equation (11)
9:     Update the scheduler from $s_{\phi_{t-2}}$ to $s_{\phi_{t-1}}$ according to Equation (7)
10:     ▽ Data Scheduling
11:     Select $\bar{\mathcal{C}}_t$ from $\mathcal{C}_t$ according to Equation (8)
12:     Update $\pi_{\theta_{t-1}}$ to $\pi_{\theta_t}$ according to Equation (2)
13:     ▽ Arm Representation Update
14:     Update representations associated to the data in $\mathcal{C}_t$ if needed
15: **end for**

The policy optimization function of DAPO can be considered as the same Equation (1) variant of GRPO.

**Group Sequence Policy Optimization (GSPO)** adopts the sequence-level clip rather than the token-level clip, specifically, it optimizes the policy by maximizing

$$\mathcal{J}_\theta = \mathbb{E}_{(q,a)\sim\mathcal{D},\{o_i\}_{i=1}^G\sim\pi_{\text{old}}(\cdot|q)}\left[\frac{1}{G}\sum_{i=1}^G \min\left(A_i s_i(\theta), A_i \text{clip}(s_i(\theta), 1 - \epsilon_{\text{low}}, 1 + \epsilon_{\text{high}})\right)\right], \quad (17)$$

where $s_i(\theta) = \exp\left(\frac{1}{|o_i|}\sum_{t=1}^{|o_i|}\log\frac{\pi_\theta(o_{i,t}|q,o_{i,<t})}{\pi_{\theta_{old}}(o_{i,t}|q,o_{i,<t})}\right)$ is the normalized sequence-level importance sampling weight. Equation (17) can also be written into a token form (Zheng et al., 2025a) of

$$\mathcal{J}_\theta = \mathbb{E}_{(q,a)\sim\mathcal{D},\{o_i\}_{i=1}^G\sim\pi_{\text{old}}(\cdot|q)}\left[\frac{1}{G}\sum_{i=1}^G\frac{1}{|o_i^t|}\sum_{t=1}^{|o_i^t|}\min(A_i s_{i,t}(\theta), A_i\text{clip}(s_{i,t}(\theta), 1 - \epsilon_{\min}, 1 + \epsilon_{\max}))\right], \quad (18)$$

where $s_{i,t}(\theta) = \frac{\text{sg}[s_i(\theta)]*\pi_{old}(y_{i,t}|x_i,y_{i,<t})}{\text{sg}(\pi_\theta(y_{i,t}|x,y_{i,<t}))} \cdot r_{i,t}(\theta)$ and sg indicates the stop gradient operation. Let $w_{i,t}$ indicates $\frac{\text{sg}[s_i(\theta)]*\pi_{old}(y_{i,t}|x,y_{i,<t})}{\text{sg}(\pi_\theta(y_{i,t}|x,y_{i,<t}))}$. The GSPO can be considered as a variant of Equation (1) by setting $\Phi$ to

$$\Phi(A, r_{i,t}(\theta), \epsilon_{\text{low}}, \epsilon_{\text{high})}) = \min(A_i \cdot w_{i,t} \cdot r_{i,t}(\theta), A_i\text{clip}(w_{i,t} \cdot r_{i,t}(\theta), 1 - \epsilon_{\text{low}}, 1 + \epsilon_{\text{high}})). \quad (19)$$

### A.2. Training Pipeline

We provide a visual illustration for the RLVR training pipeline in Figure 5.

## B. Proof of Theorems

### B.1. Definitions

We instantiate the selector network $s_\phi(\cdot)$ as a fully connected network with depth $L \geq 2$ and width $m$, where $s_\phi(h) = \mathbf{W}_L\sigma(\mathbf{W}_{L-1}\sigma(\mathbf{W}_{L-2}\cdots\sigma(\mathbf{W}_1 h)))$ with $\sigma(\cdot)$ being ReLU activation, $\mathbf{W}_1 \in \mathbb{R}^{m\times 10}$, $\mathbf{W}_i \in \mathbb{R}^{m\times m}$ for $1 \leq i \leq L-1$, $\mathbf{W}_L \in \mathbb{R}^{1\times m}$, and $\phi = [\text{vec}(\mathbf{W}_L), \ldots, \text{vec}(\mathbf{W}_1)]$. For initialization, entries of $\mathbf{W}_L$ are drawn independently from $\mathcal{N}(0, 1/m)$, and for $1 \leq i \leq L-1$, entries of each $\mathbf{W}_i$ are drawn independently from $\mathcal{N}(0, 2/m)$. We slightly abuse the notations and rewrite the arm representations $\{h_i^t\}_{i\in[M],t\in[T]}$ and rewards $\{r_i^t\}_{t\in[T],i\in[M]}$ into $\{h_i\}_{i\in[TM]}$ and $\{r_i\}_{i\in[TM]}$ respectively. The NTK matrix is defined as below,

**Definition B.1** (NTK (Jacot et al., 2018; Wang et al., 2021)). Let $\mathcal{N}$ denote the normal distribution. Given the data instances $\{h_t\}_{t=1}^{TM}$, for all $i, j \in [TM]$, define:

$$H_{i,j}^0 = \Sigma_{i,j}^0 = \langle h_i, h_j \rangle, \quad A_{i,j}^l = \begin{pmatrix} \Sigma_{i,i}^l & \Sigma_{i,j}^l \\ \Sigma_{j,i}^l & \Sigma_{j,j}^l \end{pmatrix},$$

$$\Sigma_{i,j}^l = 2\mathbb{E}_{a,b \sim \mathcal{N}(0, A_{i,j}^{l-1})}[\sigma(a)\sigma(b)],$$

$$H_{i,j}^l = 2H_{i,j}^{l-1}\mathbb{E}_{a,b \sim \mathcal{N}(0, A_{i,j}^{l-1})}[\sigma'(a)\sigma'(b)] + \Sigma_{i,j}^l.$$

Then the NTK matrix is defined as $H = (H^L + \Sigma^L)/2$

## B.2. Lemma on Bounded Deviation

Given parameters $\theta_{t-1}$ and $\theta_t$, the terms $V_t$ and $V_{t+1}$ can be viewed as stochastic estimates of the expected performance of $\theta_{t-1}$ and $\theta_t$ on the training set $\mathcal{D}$ (and similarly for $E_t$ and $E_{t+1}$). The following lemma shows that the deviation between the stochastic estimate and its corresponding expectation is bounded.

**Lemma B.2.** Let $R(\overline{\mathcal{C}}_t, \theta_{t-1}, \theta_t)$ denote the reward in (5). Assume the per-sample reward satisfies $v_i^t \in \{0, 1\}$ for all $t, i$, and let $N = |\mathcal{V}|$ be the vocabulary size. Then

$$\left| \mathbb{E}\left[ R(\overline{\mathcal{C}}_t, \theta_{t-1}, \theta_t) \right] - R(\overline{\mathcal{C}}_t, \theta_{t-1}, \theta_t) \right| \leq 2\left(1 + w_e \log N\right), \tag{20}$$

*Proof.* Since $v_i^t \in \{0, 1\}$, we have $0 \leq V_t = \text{mean}(\{v_i^t\}) \leq 1$, hence

$$V_{t+1} - V_t \in [-1, 1]. \tag{21}$$

For the entropy term, for any context the Shannon entropy of a distribution over a vocabulary of size $N$ is bounded by

$$0 \leq H(\pi_{\theta_{t-1}}(\cdot \mid \cdot)) \leq \log N. \tag{22}$$

By definition, $e_i^t$ is the average of token-level entropies, so $e_i^t \in [0, \log N]$, and thus

$$E_t = \text{mean}(\{e_i^t\}) \in [0, \log N], \qquad E_{t+1} - E_t \in [-\log N, \log N]. \tag{23}$$

Moreover, $\mathbb{I}[E_t > e_{\min}] \in \{0, 1\}$, so

$$-w_e \mathbb{I}[E_t > e_{\min}] (E_{t+1} - E_t) \in [-w_e \log N, \ w_e \log N]. \tag{24}$$

Therefore,

$$R(\overline{\mathcal{C}}_t, \theta_{t-1}, \theta_t) \in \left[ -(1 + w_e \log N), \ 1 + w_e \log N \right]. \tag{25}$$

For any random variable $X \in [a, b]$, we have $|\mathbb{E}[X] - X| \leq b - a$. Applying this with $a = -(1 + w_e \log N)$ and $b = 1 + w_e \log N$ yields

$$|\mathbb{E}[R] - R| \leq 2(1 + w_e \log N), \tag{26}$$

which proves the claim. $\qquad\square$

## B.3. Lemma for Neural Regression

**Lemma B.3** (Theorem 3.1 of (Yun et al., 2019)). *Given any dataset $\{(x_i, y_i)\}_{i=1}^N$ that all $x_i$ are different and all $y_i \in [-1, 1]$. If $L >= 3$ and $4 \lfloor \frac{m}{2} \rfloor \lfloor \frac{m}{2} \rfloor \geq N$, there exist a $\phi'$ such that $s_{\phi'}(x_i) = y_i$ for all $i \in [N]$.*

**Lemma B.4** (Lemma B.1 of (Ban et al., 2023)). *Let $\mathcal{H} = \{(h_j, r_j)\}_{j=1}^N$ be a finite feature-label set satisfying $\|h_i\|_2 = 1$ and $r_i \in [0, 1]$ for any $i \in [N]$. For any $R > 0$ and $\delta \in (0, 1)$, if $m \geq \Omega\left(\text{poly}\left(R, L, N, \log(1/\delta)\right)\right)$, then with probability at least $1 - \delta$ over the initialization $\phi_0$, the following bounds hold uniformly for all $(h, r) \in \mathcal{H}$ and all $\phi \in B_R(\phi_0) = \{\phi : \|\phi - \phi_0\| \leq R/m^{1/4}\}$:*

$$|s_\phi(h)| \leq O(1), \qquad \|\nabla_\phi s_\phi(h)\|_2 \leq O(\sqrt{L}), \qquad \|\nabla_\phi \ell(\phi; h, r)\|_2 \leq O(\sqrt{L}) \tag{27}$$

**Lemma B.5** (Uniform local perturbation bound). *Let $\mathcal{H} = \{(h_j, r_j)\}_{j=1}^N$ be a finite feature-label set satisfying $\|h_i\|_2 = 1$ and $r_i \in [0, 1]$ for any $i \in [N]$, and let $\ell(\phi; h, r) = \frac{1}{2}(s_\phi(h) - r)^2$. For any $R > 0$ and $\delta \in (0, 1)$, if $m \geq \Omega(\text{poly}(R, L, N, \log(1/\delta)))$, then with probability at least $1 - \delta$ over the initialization $\phi_0$, the following bounds hold uniformly for all $(h, r) \in \mathcal{H}$ and all $\phi, \phi' \in B_R(\phi_0) = \{\phi : \|\phi - \phi_0\| \leq R/m^{1/4}\}$:*

$$|s_\phi(h) - s_{\phi'}(h)| \leq O\left(\frac{R\sqrt{L}}{m^{1/4}}\right). \tag{28}$$

*Proof.* Let $\rho = R/m^{1/4}$ and choose $\omega_\star = c_0 L^{-9/2}(\log m)^{-3}$, where $c_0 > 0$ is a sufficiently small universal constant. For $m \geq \Omega(\text{poly}(R, L, N, \log(1/\delta)))$, we have $\rho \leq \omega_\star$ and $O(NL^2)\exp(-\Omega(m\omega_\star^{2/3}L)) \leq \delta/2$. Hence, $B_R(\phi_0)$ is contained in the local ball of radius $\omega_\star$. By Lemma B.2 of (Ban et al., 2023), uniformly for all $(h, r) \in \mathcal{H}$ and all $\phi, \phi' \in B_R(\phi_0)$,

$$|s_\phi(h) - s_{\phi'}(h) - \langle \nabla_\phi s_{\phi'}(h), \phi - \phi'\rangle| \leq O(\omega_\star^{1/3} L^2 \sqrt{\log m})\|\phi - \phi'\|_2.$$

Since $\omega_\star^{1/3} L^2 \sqrt{\log m} \leq O(\sqrt{L})$, this gives

$$|s_\phi(h) - s_{\phi'}(h) - \langle \nabla_\phi s_{\phi'}(h), \phi - \phi'\rangle| \leq O(\sqrt{L})\|\phi - \phi'\|_2.$$

By Lemma B.4, $\|\nabla_\phi s_{\phi'}(h)\|_2 \leq O(\sqrt{L})$ with probability at least $1 - \delta/2$. Taking a union bound over the two events, with probability at least $1 - \delta$,

$$|s_\phi(h) - s_{\phi'}(h)| \leq |\langle \nabla_\phi s_{\phi'}(h), \phi - \phi'\rangle| + O(\sqrt{L})\|\phi - \phi'\|_2$$

$$\leq O(\sqrt{L})\|\phi - \phi'\|_2 \leq O\left(\frac{R\sqrt{L}}{m^{1/4}}\right),$$

where the last step uses $\|\phi - \phi'\|_2 \leq 2R/m^{1/4}$. $\qquad\qquad\square$

**Lemma B.6** (Lemma 5.2 of (Ban et al., 2023)). *Let $\mathcal{H} = \{(h_j, r_j)\}_{j=1}^N$ be a finite feature-label set satisfying $\|h_i\|_2 = 1$ and $r_i \in [0, 1]$ for any $i \in [N]$. For any arm representations $\{h_i\}_{i\in[TM]}$ and rewards $\{r_i\}_{i\in[TM]}$ sampled from $\mathcal{H}$ and satisfying Assumption 5.2, if $m \geq \Omega\left(\text{poly}(T, R, L, N, \log(\delta), \frac{1}{\lambda_0})\right)$, with probability at least $1 - \delta$ over initialization of $\phi_0$, there exists $\phi' \in B(\phi_0, \tilde{\Omega}(T^{3/2}))$ such that*

$$\sum_{t=1}^{TM} \left(s_{\phi'}(h_t) - r_t\right)^2 \leq \tilde{\mathcal{O}}\left(\sqrt{\tilde{d}} + S\right)^2 * \tilde{d}, \tag{29}$$

*where $S = \sqrt{r^T H^{-1} r}$ and $\tilde{d} = \frac{\log \det(I+H)}{\log(1+TM)}$.*

## B.4. Proof of Theorem 5.3

Let $\phi^*$ denote the optimal policy parameter for buffer size $jBG$. To prove Theorem 5.3, we can show that there exists a policy parameter $\phi'$ such that, for any given rollout sequence $\mathcal{R} = \{\mathcal{R}_1, \ldots, \mathcal{R}_T\}$, the selected data sequence $x_1(\phi', \mathcal{R}, iBG), \ldots, x_T(\phi', \mathcal{R}, iBG)$ is identical to the sequence $x_1(\phi^*, \mathcal{R}, jBG), \ldots, x_T(\phi^*, \mathcal{R}, jBG)$, where $x_t(\phi, \mathcal{R}, M)$ represents the data chosen by scheduler $s_\phi$ when the buffer size is $M$ and the rollout data is $\mathcal{R}$. Therefore, we can write the following inequality:

$$\varphi(iBG) = \max_\phi \mathbb{E}_{M=iBG}[\max_{t\in T} g(\theta_t)] \geq \mathbb{E}_{\phi=\phi', M=iBG}[\max_{t\in T} g(\theta_t)] = \varphi(jBG),$$

where $\mathbb{E}_{\phi=\phi', M=iBG}[\max_{t\in T} g(\theta_t)]$ represents the optimal performance when the buffer size is $iBG$ and the scheduler parameter is $\phi'$.

Next, we describe how to construct $\phi'$ based on $\phi^*$. Since the vocabulary size $|V|$, the response length $L_{\max}$, and the number of training rounds $T$ are finite, all possible rollout data and arm representations come from a finite set. Let $H \subset \mathbb{R}^{10}$ denote

the set of all possible arm representations, and let $U = \max_{h \in H} |s_{\phi^*}(h)|$ be the upper bound of the absolute value of scores, with $C = |H|$ representing the size of the arm set. We assign a label $y_h$ for each $h \in H$ as follows:

$$y_h = \frac{2U \cdot \mathbb{I}[h_s < j] + s_{\phi^*}(h)}{3U},$$

where $h_s$ denotes the delta step feature for the arm representations in Table 1. By setting $L = 3$, $m = 2C$, and applying Lemma B.3, we obtain $s_{\phi'}(h) = y_h$ for all $h \in H$.

Now, consider the scores $s_{\phi'}(h), s_{\phi'}(h')$ for any two data points $x, x' \in \mathcal{F}_t$, where $h, h'$ are the representations for $x$ and $x'$, respectively. The buffer $\mathcal{F}_t = \cup_{i=t-i+1}^{t} \mathcal{R}_t$ includes the data from the latest $i$ steps. For data within the latest $j$ steps, the delta step $h_s < j$, so $y_h$ will be larger than the labels for data older than the latest $j$ steps according to the definition of $y_h$. For two data points both within the latest $j$ steps, we have:

$$s_{\phi'}(h) - s_{\phi'}(h') = \frac{s_{\phi^*}(h) - s_{\phi^*}(h')}{3U}.$$

Thus, the relative difference in scores is the same as for $s_{\phi^*}$. Combining the two situations together, the strategy $s_{\phi'}$ behaves identically to $s_{\phi^*}$. As a result, for any rollout sequence $\mathcal{R} = \{\mathcal{R}_1, \ldots, \mathcal{R}_T\}$, the data selection sequence of $s_{\phi'}$ will be the same as that of $s_{\phi^*}$. This completes the proof.

### B.5. Proof of Theorem 5.4

*Proof.* For notation simplicity, write $g_a^t = g(\mathcal{A}(x_a^t, \theta_{t-1}))$. Let $a_t^* = \arg\max_{a \in [M]} g_a^t$, $a_t^g = \arg\max_{a \in [M]} s_{\phi_{t-1}}(h_a^t)$, and let $a_t^e$ denote the exploration arm. Let $e_t \sim \text{Bernoulli}(\epsilon_t)$ indicate whether exploration is performed, so $a_t = a_t^e$ if $e_t = 1$ and $a_t = a_t^g$ otherwise. Then

$$\mathbb{E}[R_T] = \sum_{t=1}^{T} \mathbb{E}\left[g_{a_t^*}^t - g_{a_t}^t\right]$$

$$= \sum_{t=1}^{T} \mathbb{E}\left[g_{a_t^*}^t - g_{a_t^g}^t\right] + \sum_{t=1}^{T} \mathbb{E}\left[e_t\left(g_{a_t^g}^t - g_{a_t^e}^t\right)\right]$$

$$\overset{(a)}{\leq} \sum_{t=1}^{T} \mathbb{E}\left[g_{a_t^*}^t - g_{a_t^g}^t\right] + \sum_{t=1}^{T} \mathbb{E}\left[e_t\right]$$

$$\leq \sum_{t=1}^{T} \mathbb{E}\left[g_{a_t^*}^t - g_{a_t^g}^t\right] + \sum_{i=1}^{T} \epsilon_t,$$

where (a) holds because the reward range is $[0,1]$. By setting $T_w = 1$, $\epsilon_0 = 1$, $\Delta_\epsilon = T^{-1/2}$, and $\epsilon_{\min} = 0$, the exploration probability in Equation (9) becomes $\epsilon_1 = 1$ and, for $t \geq 2$, $\epsilon_t = \max(1 - (t-1)/\sqrt{T}, 0)$. Therefore,

$$\sum_{t=1}^{T} \epsilon_t = 1 + \sum_{t=2}^{T} \max\left(1 - \frac{t-1}{\sqrt{T}}, 0\right)$$

$$= 1 + \sum_{k=1}^{T-1} \max\left(1 - \frac{k}{\sqrt{T}}, 0\right),$$

where $k = t - 1$. The summand is positive only when $k < \sqrt{T}$, so there are at most $\lceil\sqrt{T}\rceil - 1$ nonzero terms in the second summation. Since each summand is at most 1,

$$\sum_{t=1}^{T} \epsilon_t \leq 1 + (\lceil\sqrt{T}\rceil - 1) \leq 1 + \sqrt{T} \leq 2\sqrt{T}.$$

Using $s_{\phi_{t-1}}(h^t_{a^g_t}) \geq s_{\phi_{t-1}}(h^t_{a^*_t})$, for any auxiliary parameter sequence $\{\phi^*_t\}^T_{t=0}$, we have

$$\sum_{t=1}^{T} \mathbb{E}\left[g^t_{a^*_t} - g^t_{a^g_t}\right] \overset{(a)}{=} \sum_{t=1}^{T} \mathbb{E}\left[r^t_{a^*_t} - r^t_{a^g_t}\right]$$

$$\leq \underbrace{\sum_{t=1}^{T} \mathbb{E}\left[\left|r^t_{a^*_t} - s^*_{\phi_{t-1}}(h^t_{a^*_t})\right|\right]}_{I_1} + \underbrace{\sum_{t=1}^{T} \mathbb{E}\left[\left|s_{\phi^*_{t-1}}(h^t_{a^*_t}) - s_{\phi_{t-1}}(h^t_{a^*_t})\right|\right]}_{I_2} + \underbrace{\sum_{t=1}^{T} \mathbb{E}\left[\left|s_{\phi_{t-1}}(h^t_{a^g_t}) - g^t_{a^g_t}\right|\right]}_{I_3},$$

where $(a)$ holds because $g^t_a = \mathbb{E}[r^t_a \mid \mathcal{G}_t]$, where $\mathcal{G}_t$ denotes the information available before observing the round-$t$ rewards. By the law of iterated expectation and the fact that $a^*_t$ and $a^g_t$ are determined by $\mathcal{G}_t$, we have $\mathbb{E}[r^t_{a^*_t} - r^t_{a^g_t}] = \mathbb{E}[g^t_{a^*_t} - g^t_{a^g_t}]$.

We now define the auxiliary trajectory used in $I_1$ and $I_2$. Let $\phi^*_0 = \phi_0$ and $\ell(\phi, h, r) = \frac{1}{2}(s_\phi(h) - r)^2$, we define $\phi^*_t = \phi^*_{t-1} - \eta_2 \nabla_\phi \ell(\phi^*_{t-1}, h^t_{a^*_t}, r^t_{a^*_t})$. The actual scheduler trajectory can be written as $\phi_t = \phi_{t-1} - \eta_2 \nabla_\phi \ell(\phi_{t-1}; h^{\mathrm{obs}}_t, r^{\mathrm{obs}}_t)$ for some observed feature-label pair $(h^{\mathrm{obs}}_t, r^{\mathrm{obs}}_t)$. Since the initial checkpoint, training set, vocabulary, maximum response length, and training horizon are fixed, and the rollout randomness has finite support over $T$ rounds, all possible scheduler feature-label pairs are sampled from a finite set $\mathcal{H}_{\mathrm{all}} = \{(h_j, r_j)\}^{N_{\mathcal{H}}}_{j=1}$ with $\|h_j\|_2 = 1$ and $r_j \in [0, 1]$. Let $\mathcal{E}_1$, $\mathcal{E}_2$, and $\mathcal{E}_3$ be the events in Lemma B.4, Lemma B.5, and Lemma B.6 with failure probabilities $\delta_1$, $\delta_2$, and $\delta_3$, respectively. Set $\delta_1 = \delta_2 = \delta_3 = \delta/3$, $R = \widetilde{\Omega}(T^{3/2})$, and $\eta_2 = R^2/\sqrt{m}$. By a union bound, $\mathcal{E} = \mathcal{E}_1 \cap \mathcal{E}_2 \cap \mathcal{E}_3$ holds with probability at least $1 - \delta$ over $\phi_0$. In the following, we condition on $\mathcal{E}$.

On $\mathcal{E}_1$, both $\{\phi_t\}^T_{t=0}$ and $\{\phi^*_t\}^T_{t=0}$ stay in $B_R(\phi_0)$. Indeed, by Lemma B.4, $\|\nabla_\phi \ell(\phi; h, r)\|_2 \leq O(\sqrt{L})$ uniformly over $(h, r) \in \mathcal{H}_{\mathrm{all}}$ and $\phi \in B_R(\phi_0)$. Therefore, for either trajectory,

$$\|\phi_t - \phi_0\|_2 \leq \sum_{\tau=1}^{T} \eta_2 O(\sqrt{L}) \leq O\left(\frac{TR^2\sqrt{L}}{\sqrt{m}}\right) \leq \frac{R}{m^{1/4}},$$

provided that $m \geq \Omega(T^4 R^4 L^2)$. The same argument applies to $\phi^*_t$. On $\mathcal{E}_2$, by Lemma B.6 and the choice $R = \widetilde{\Omega}(T^{3/2})$, there exists $\bar{\phi} \in B_R(\phi_0)$ such that

$$\mathcal{Q}_T := \sum_{t=1}^{T} \sum_{a=1}^{M} \left(s_{\bar{\phi}}(h^t_a) - r^t_a\right)^2 \leq \widetilde{O}\left(\sqrt{\widetilde{d}} + S\right)^2 \widetilde{d}.$$

Finally, on $\mathcal{E}_2$, for any $\phi, \phi' \in B_R(\phi_0)$ and any $(h, r) \in \mathcal{H}_{\mathrm{all}}$,

$$|s_\phi(h) - s_{\phi'}(h)| \leq \Delta_m, \qquad \Delta_m = O\left(\frac{R\sqrt{L}}{m^{1/4}}\right).$$

The same lower bound $m \geq \Omega(T^4 R^4 L^2)$ also gives $T\Delta_m = O(1)$.

We now bound the three terms. For $I_3$,

$$I_3 = \sum_{t=1}^{T} \mathbb{E}\left[\left|s_{\phi_{t-1}}(h^t_{a^g_t}) - s_{\bar{\phi}}(h^t_{a^g_t}) + s_{\bar{\phi}}(h^t_{a^g_t}) - r^t_{a^g_t}\right|\right]$$

$$\leq \sum_{t=1}^{T} \mathbb{E}\left[\left|s_{\bar{\phi}}(h^t_{a^g_t}) - r^t_{a^g_t}\right|\right] + T\Delta_m$$

$$\overset{(a)}{\leq} \sqrt{T}\mathbb{E}\left[\sqrt{\sum_{t=1}^{T}\left(s_{\bar{\phi}}(h^t_{a^g_t}) - r^t_{a^g_t}\right)^2}\right] + O(1)$$

$$\leq \sqrt{T}\mathbb{E}\left[\sqrt{\mathcal{Q}_T}\right] + O(1)$$

$$\leq \mathbb{E}\left[\widetilde{O}\left(\sqrt{T}\left(\widetilde{d} + S\sqrt{\widetilde{d}}\right)\right)\right] + O(1),$$

where $(a)$ comes by applying the Cauchy–Schwarz inequality. The same argument gives the bound for $I_1$,

$$
\begin{aligned}
I_1 &\le \sum_{t=1}^{T} \mathbb{E}\left[\left|s_{\tilde{\phi}}(h_{a_t^*}^t) - r_{a_t^*}^t\right|\right] + T\Delta_m \\
&\le \sqrt{T}\mathbb{E}\left[\sqrt{\mathcal{Q}_T}\right] + O(1) \\
&\le \mathbb{E}\left[\widetilde{O}\left(\sqrt{T}\left(\tilde{d} + S\sqrt{\tilde{d}}\right)\right)\right] + O(1).
\end{aligned}
$$

For $I_2$, since both $\phi_{t-1}^*$ and $\phi_{t-1}$ lie in $B_R(\phi_0)$, Lemma B.5 directly yields $I_2 \le \sum_{t=1}^{T} \Delta_m = O(1)$.

Combining the bounds above, if $m \ge \Omega(\mathrm{poly}(N_{\mathcal{H}}, T, R, L, \log(\frac{1}{\delta}), \frac{1}{\lambda_0}))$, with probability at least $1 - \delta$ over the scheduler initialization, we have

$$
\begin{aligned}
\mathbb{E}[R_T] &\le I_1 + I_2 + I_3 + \sum_{t=1}^{T} \epsilon_t \\
&\le \mathbb{E}\left[\widetilde{O}\left(\sqrt{T}\left(\tilde{d} + S\sqrt{\tilde{d}}\right) + \sqrt{T}\right)\right].
\end{aligned}
$$

This completes the proof. $\qquad\square$

## C. Discussion on Subset Selection

We first justify the subset selection as sample scoring using a first-order Taylor expansion, and then discuss the regret bound under subset selection.

**Justification of Sample Selection.** When $\theta_{t-1}$ is fixed, the expression

$$
R(\bar{\mathcal{C}}_t, \theta_{t-1}, \theta_t) = V_{t+1} - V_t - w_e\mathbb{I}[E_t > e_{\min}](E_{t+1} - E_t) \tag{30}
$$

can be interpreted as a stochastic estimate of the function $f(\theta_t)$, which is defined as

$$
f(\theta_t) = \mathbb{E}_{(q,a)\sim D, o\sim \pi_{\theta_t}(\cdot|q)}\left[v(a,o) - V_t - w_e\mathbb{I}[E_t > e_{\min}] \cdot \left(\sum_{t=1}^{|o|} \frac{-\sum_{x\in\mathcal{V}} \pi_{\theta_t}(x|q,o_{<t})\log\pi_{\theta_t}(x|q,o_{<t})}{|o|} - E_t\right)\right], \tag{31}
$$

where $v(\cdot)$ is the reward function of RLVR. Applying the first-order Taylor expansion around the reference point $\theta_{t-1}$, we will get

$$
f(\theta_t) \approx f(\theta_{t-1}) + \nabla_{\theta_{t-1}} f(\theta) \cdot (\theta_t - \theta_{t-1}). \tag{32}
$$

Since the parameter is updated by gradient ascent, i.e., $\theta_t = \theta_{t-1} + \eta\nabla_{\theta_{t-1}} J_\theta(\bar{C}_t)$, we obtain

$$
\begin{aligned}
f(\theta_t) &\approx f(\theta_{t-1}) + \nabla_{\theta_{t-1}} f(\theta) \cdot \eta\nabla_{\theta_{t-1}} J_\theta(\bar{C}_t) \\
&= f(\theta_{t-1}) + \nabla_{\theta_{t-1}} f(\theta) \cdot \eta \sum_{x\in\bar{C}_t} \nabla_{\theta_{t-1}} J_\theta(\bar{C}_t).
\end{aligned} \tag{33}
$$

Therefore, the subset utility $f(\theta_t)$ becomes the sum of the utilities of the samples.

**Regret bound under subset selection.** Under the first-order approximation, the subset-level reward is additive. If we treat the advantage-dispatched per-sample reward as the (unbiased) true reward, the resulting problem can be modeled as a contextual semi-bandit. In each round, selecting a size-$K$ subset can be viewed as making $K$ single-arm choices (with distinct arms) and receiving feedback for all selected arms at the end of the round. Therefore, any adversarial single-play regret guarantee extends to the subset setting with at most a factor-$K$ degradation, i.e., the cumulative regret is upper bounded by $K$ times the single-rollout selection bound (up to the same logarithmic factors).

| Experiment | $M$ | $\epsilon_0$ | $\eta_2$ | $T_w$ | $\alpha$ | $\epsilon_{\min}$ | $\delta$ | $p\%$ | $e_{\min}$ |
|---|---|---|---|---|---|---|---|---|---|
| **Qwen3-1.7B-Base** | | | | | | | | | |
| GRPO + CBS | 2048 | 1 | 0.0001 | 50 | 0.9 | 0.2 | 0.008 | - | 0.1 |
| GRPO + CBS* | 1024 | 1 | 0.0001 | 50 | 0.9 | 0.2 | 0.008 | 30% | 0.1 |
| DAPO + CBS | 2048 | 1 | 0.0001 | 50 | 0.9 | 0.2 | 0.008 | - | 0.5 |
| DAPO + CBS* | 1024 | 1 | 0.0001 | 50 | 0.9 | 0.2 | 0.008 | 30% | 0.5 |
| GSPO + CBS | 2048 | 1 | 0.0001 | 50 | 0.9 | 0.005 | 0.008 | - | 0.5 |
| GSPO + CBS* | 1024 | 1 | 0.0001 | 50 | 0.9 | 0.005 | 0.008 | 30% | 0.5 |
| **Qwen3-4B-Base** | | | | | | | | | |
| GRPO + CBS | 2048 | 1 | 0.0001 | 50 | 0.9 | 0.2 | 0.008 | - | 0.1 |
| GRPO + CBS* | 1024 | 1 | 0.0001 | 50 | 0.9 | 0.2 | 0.008 | 30% | 0.1 |
| DAPO + CBS | 2048 | 1 | 0.0001 | 50 | 0.9 | 0.2 | 0.008 | - | 0.5 |
| DAPO + CBS* | 1024 | 1 | 0.0001 | 50 | 0.9 | 0.2 | 0.008 | 30% | 0.5 |
| GSPO + CBS | 2048 | 1 | 0.0001 | 0 | 0.9 | 0.005 | 0.008 | - | 0.5 |
| GSPO + CBS* | 1024 | 1 | 0.0001 | 50 | 0.9 | 0.005 | 0.008 | 30% | 0.5 |
| **Qwen3-8B-Base** | | | | | | | | | |
| GRPO + CBS | 2048 | 1 | 0.0001 | 50 | 0.9 | 0.2 | 0.008 | - | 0.1 |
| GRPO + CBS* | 1024 | 1 | 0.0001 | 50 | 0.9 | 0.2 | 0.008 | 30% | 0.1 |
| DAPO + CBS | 2048 | 1 | 0.0001 | 50 | 0.9 | 0.2 | 0.008 | - | 0.8 |
| DAPO + CBS* | 1024 | 1 | 0.0001 | 50 | 0.9 | 0.2 | 0.008 | 30% | 0.5 |
| GSPO + CBS | 2048 | 1 | 0.0001 | 50 | 0.9 | 0.005 | 0.008 | - | 0.5 |
| GSPO + CBS* | 1024 | 1 | 0.0001 | 50 | 0.9 | 0.005 | 0.008 | 30% | 0.5 |

*Table 4.* Scheduler hyperparameters (adjusted for different models and sizes).

# D. Implementation Details

**Hyperparameters.** We set the learning rate to $1 \times 10^{-6}$, the batch size to 128, the mini-batch size to 16, and the rollout number per prompt to 8. For the scheduler, we set the buffer size to 2048 for global scheduling and $p$ to 30% for intra-group scheduling. We keep the mini-batch size unchanged, and the policy optimization step decreases when rollouts are downsampled by the online scheduler. For example, if we generate 1024 rollouts and the scheduler downsamples them to $\lfloor 1024 \times 0.3 \rfloor = 307$, we then rearrange these rollouts into $\lfloor 307/(16 \times 8) \rfloor = 2$ mini-batches with each having 128 rollouts. The entropy penalty weight $w_e$ is set to 100. In our implementation, we approximate intra-group selection by pooling rollouts from all groups generated at the same step and selecting the top-$p$% globally, which yields a comparable retained set while simplifying the scheduler. More method-specific hyperparameters are summarized in Table 4. Following (Liu et al., 2025a), we use the prompt template below,

---

**Prompt Template**

```
<|im_start|>system
You are a helpful assistant.<|im_end|>
<|im_start|>user
{Problem}
Please reason step by step, and put your final answer within boxed{}.<|im_end|>
<|im_start|>assistant
```

---

**Train and evaluation.** We control the rollout volume during training to ensure a fair comparison across different policy optimization methods. Specifically, we train GRPO and GSPO for 500 steps. For DAPO, its dynamic sampling requires first upsampling the rollout by a factor of three, followed by downsampling the non-zero advantage ones. Therefore, we train DAPO for 200 steps. During training, we evaluate every 20 steps, reporting Avg@1 for larger datasets (i.e., MATH500, Minerva, and Olympiad) and Avg@16 for smaller datasets (i.e., AIME24 and AIME25). After training, we use the checkpoint with the highest average evaluation score to perform evaluation on the same dataset, computing Avg@4 for larger datasets and Avg@32 for smaller datasets. The temperature and maximum response length are set to 1.0 and 4096, respectively, for both training and evaluation. For the reward function, we use *Math_Verify*[4] to compare the response with the answer, assigning a value of 1 if they match, and 0 otherwise.

# E. More Experimental Results

### E.1. Training Dynamics

We show the dynamics of average score on the validation set, training reward and actor update time in Figure 6, Figure 7 and Figure 8 respectively.

### E.2. Ablation Study

The dataset-specific performance of different variants is shown in Table 5. Then entropy dynamics of the three PO methods are shown in Figure 9.

### E.3. Hyperparameter Sensitivity

To explore the impact of buffer size $|C_t|$, we vary the buffer size of GRPO + CBS under the Qwen3-4B-Base setting, ranging from 1024 to 5120 in increments of 1024, and analyze both the performance and buffer-related runtime. As shown in Figure 11, CBS consistently outperforms the original GRPO within the range of [1024, 4096], demonstrating its robustness. However, when the buffer size reaches 5120, performance degrades below that of GRPO, which is expected due to factors such as increased noise from more data and the fact that the importance sampling in policy optimization is less likely to prioritize outdated data. Besides, the runtime generally increases as the buffer size increase. Moreover, the runtime typically increases with the buffer size. To optimize both performance and efficiency, we set the buffer size to 2048, balancing the tradeoff between performance and runtime.

---

[4]https://github.com/huggingface/Math-Verify

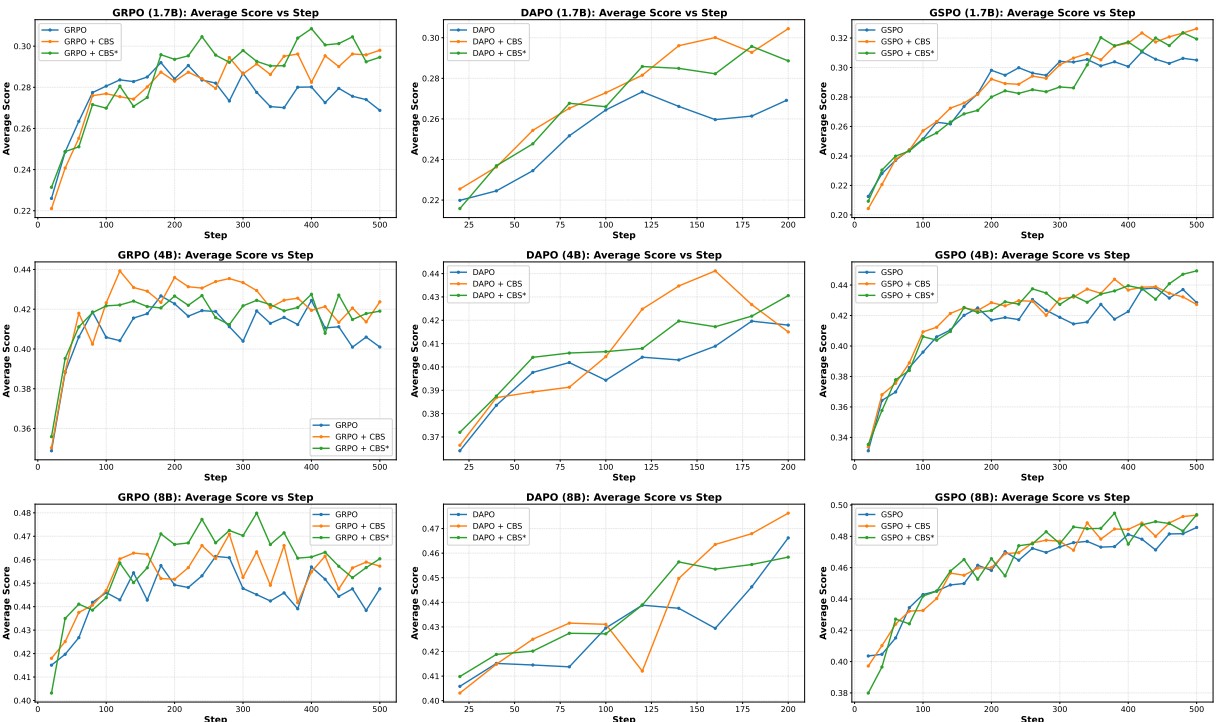

*Figure 6.* Training dynamics of the average score on the validation set

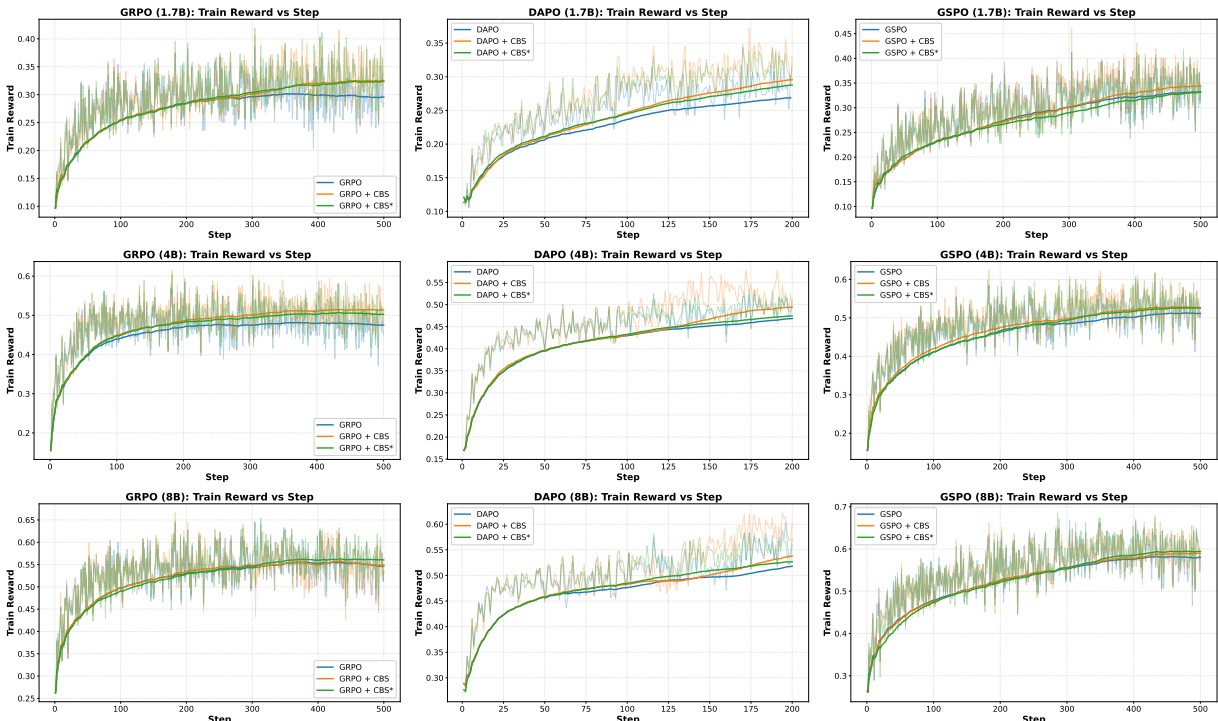

*Figure 7.* Training dynamics of the train reward.

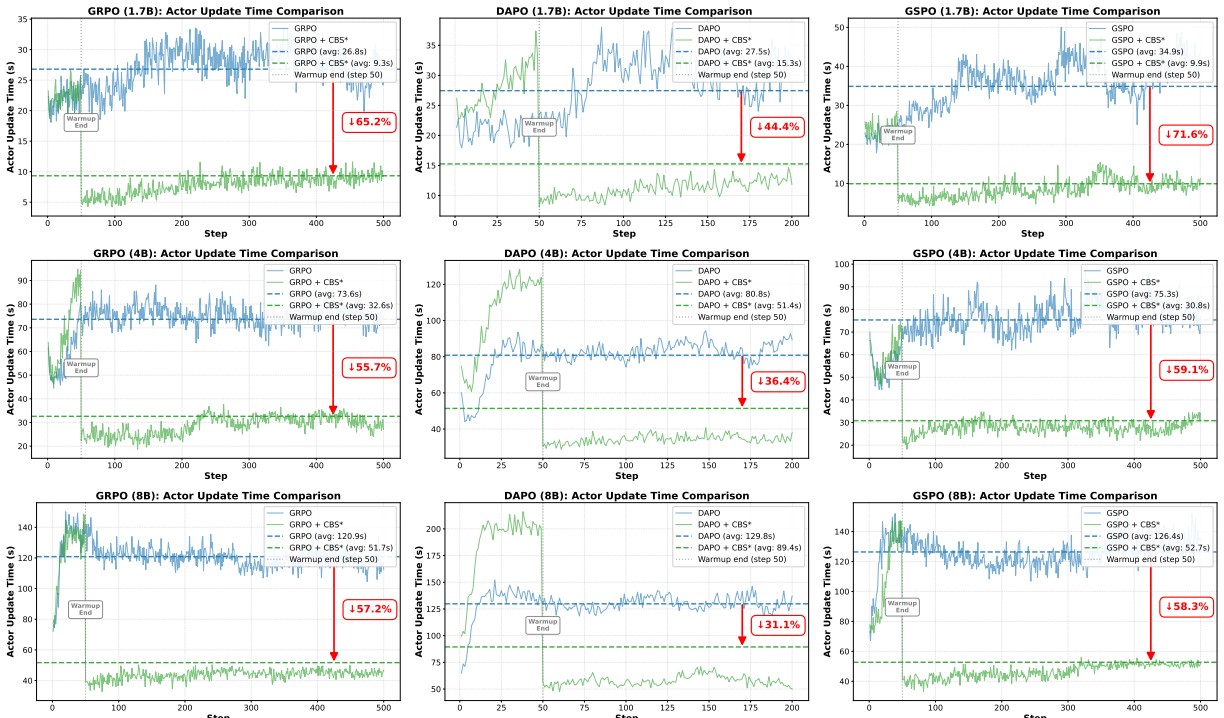

*Figure 8.* Training dynamics of the actor update time.

*Table 5.* Dataset-specific ablation study results for Qwen3-4B-Base under global scheduling.

| Variant | AIME24 | AIME25 | AMC23 | MATH500 | Minerva | Olympiad | Average |
|---|---|---|---|---|---|---|---|
| | | | | *GRPO* | | | |
| CBS | 20.31 | 18.44 | 63.05 | 74.45 | 34.47 | 43.73 | 42.41 |
| w/o EMA | 20.10 | 14.37 | 61.95 | 74.80 | 34.65 | 44.47 | 41.72 |
| w/o Entropy | 19.17 | 21.15 | 63.83 | 74.35 | 33.00 | 44.07 | **42.60** |
| Random | 17.19 | 15.83 | 61.72 | 74.85 | 35.29 | 43.58 | 41.41 |
| | | | | *DAPO* | | | |
| CBS | 20.62 | 20.94 | 62.89 | 75.85 | 36.58 | 46.70 | **43.93** |
| w/o EMA | 17.19 | 16.04 | 59.30 | 73.55 | 34.65 | 42.21 | 40.49 |
| w/o Entropy | 17.81 | 17.50 | 61.64 | 73.95 | 31.43 | 43.32 | 40.94 |
| Random | 15.94 | 17.08 | 58.44 | 73.95 | 31.07 | 41.54 | 39.67 |
| | | | | *GSPO* | | | |
| CBS | 21.46 | 18.33 | 67.03 | 76.05 | 36.49 | 44.18 | **43.92** |
| w/o EMA | 18.02 | 17.40 | 64.92 | 75.50 | 32.63 | 44.81 | 42.21 |
| w/o Entropy | 19.17 | 19.17 | 67.11 | 76.65 | 35.75 | 44.92 | 43.80 |
| Random | 16.88 | 15.21 | 61.17 | 75.15 | 32.72 | 42.14 | 40.55 |

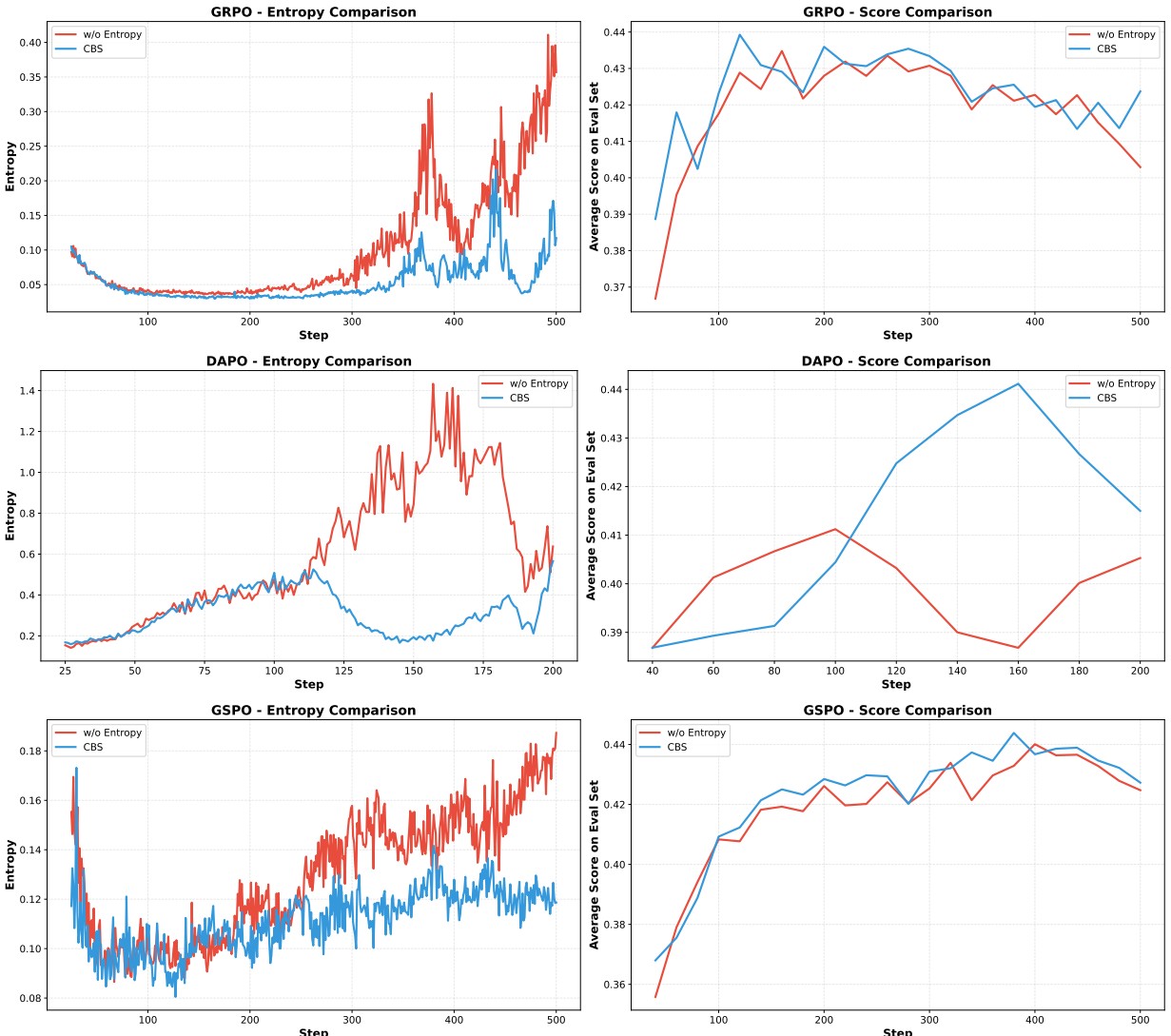

*Figure 9.* Entropy dynamics of CBS and w/o Entropy.

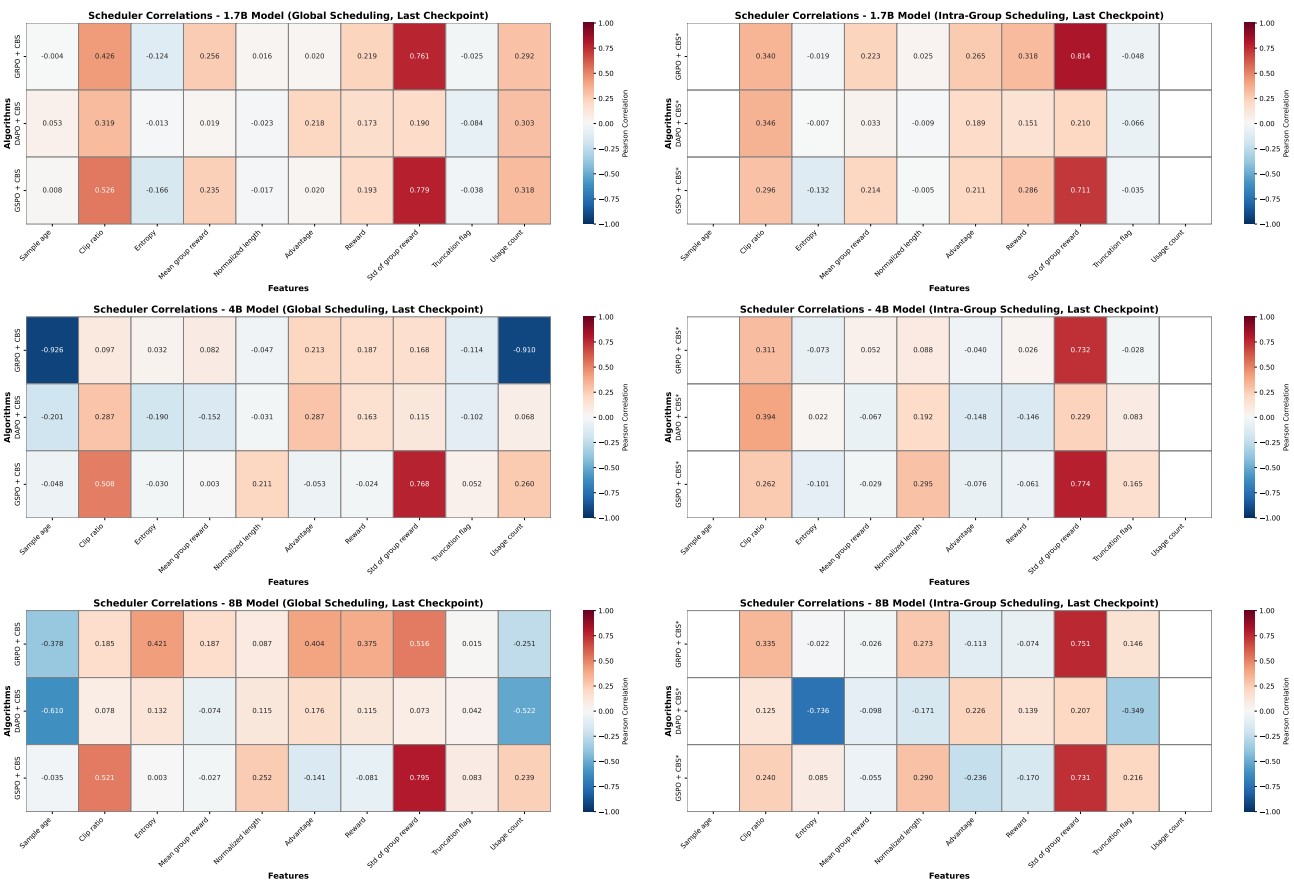

*Figure 10.* The Pearson correlation coefficient (PCC) between scheduler scores and training dynamic features, with empty blocks representing NaN values due to identical values across the training dynamic features in that dimension.

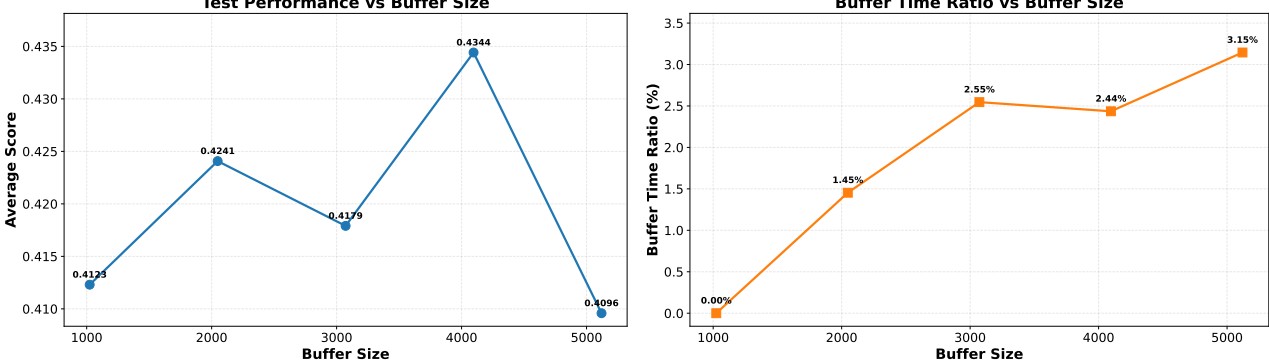

*Figure 11.* Performance and buffer-related runtime change of buffer size, where buffer size=1024 indicates the original GRPO.

### E.4. Scheduling Pattern Analysis and Case Study

We analyze the scheduling pattern of CBS by first examining the Pearson Correlation Coefficient (PCC) between training-dynamics features and the scheduler score. As shown in Figure 10, the patterns differ across policy optimization methods. For instance, GRPO + CBS relies heavily on data interaction history (i.e., usage count and sample age), while GSPO + CBS depends significantly on reward-related signals. These results demonstrate that CBS can adaptively adjust its scheduling behavior based on different policy optimization methods, leading to consistent performance improvements. Furthermore, we also respectively show rollouts that are filtered (in Figure 12) and selected by the scheduler (in Figure 13).

---

### Case Study

**Prompt:**

**system**
You are a helpful assistant.
**user**
Let $n$ be the largest integer that is the product of exactly 3 distinct prime numbers: $x$, $y$, and $10x + y$, where $x$ and $y$ are digits. What is the sum of the digits of $n$?
Please reason step by step, and put your final answer within \boxed{}.
**assistant**

**Answer**: 12

- - - - - - - - - - - - - - - - - - - - - - - - - - - - - - - - - - - - - - - - - - - - - - - - - - - -

**Model Response:**

Alright, I have this problem in front of me, and I need to find the largest integer $n$ that is the product of exactly three distinct prime numbers: $x$, $y$, and $10x + y$, where $x$ and $y$ are single-digit numbers (from 1 to 9). Then, I need to find the sum of the digits of this integer $n$.

First, let's break it down:

**Understanding the Problem**

We need three distinct prime numbers: 1. $x$: a single-digit prime number. 2. $y$: another single-digit prime number. 3. $10x + y$: a two-digit number that is also prime.
And we want the product $n = x \times y \times (10x + y)$ to be as large as possible.

**Identifying Single-Digit Prime Numbers**

Single-digit primes are: 2, 3, 5, 7. So, both $x$ and $y$ can be any of these values, and they must be different because the primes need to be distinct.

*... [Intermediate reasoning steps omitted for brevity] ...*

**Calculating the Products**

1. For $x = 2$, $y = 3$:
$$n = 2 \times 3 \times 23 = 6 \times 23 = 138$$

2. For $x = 5$, $y = 3$:
$$n = 5 \times 3 \times 53 = 15 \times 53 = 795$$

Wait, 795 is larger than 138, so it's worth keeping.
3. For $x = 3$, $y = 7$:
$$n = 3 \times 7 \times 37 = 21 \times 37 = 777$$

4. For $x = 7$, $y = 3$:
$$n = 7 \times 3 \times 73 = 21 \times 73 = 1533$$

Now, comparing all the values of $n$ we have: 138, 795, 777, 1533. The largest $n$ is 1533.

**Computing the Sum of the Digits of $n$**

Now, we need to find the sum of the digits of 1533. Let's add them:

$$1 + 5 + 3 + 3 = 12$$

**Final Answer**

Therefore, the sum of the digits of $n$ is $\boxed{12}$. NullOrUndefinedAssistant systemassistantassistantassistantassistantassistantassistantassistant

*... [Output repeats "assistant" continuously until truncate] ...*

*Figure 12.* Rollout filtered by CBS, where the model answers correctly but fails to stop, entering a repetitive loop of token generation.

---

## Case Study

**Prompt:**

**system**
You are a helpful assistant.
**user**
The positive five-digit integers that use each of the digits 1, 2, 3, 4 and 5 exactly once are ordered from least to greatest. What is the 50[th] integer in the list?
Please reason step by step, and put your final answer within \boxed{}.
**assistant**

### Answer: 31254

- - - - - - - - - - - - - - - - - - - - - - - - - - - - - - - - - - - - - - - - - - - - -

**Model Response:**

Alright, I have this problem in front of me:

*The positive five-digit integers that use each of the digits 1, 2, 3, 4, and 5 exactly once are ordered from least to greatest. What is the 50th integer in the list?*

I need to find the 50th permutation of the digits 1, 2, 3, 4, and 5 arranged in ascending order. Let's break this down.

**Understanding the Problem**

First, I need to generate five-digit numbers using each digit from 1 to 5 exactly once, sorted from smallest to largest, and then find the number that is in the 50th position.

**Total Number of Permutations**

I know that the total number of unique five-digit numbers using each of the digits 1, 2, 3, 4, and 5 exactly once is the number of permutations of 5 distinct digits.

The number of permutations of 5 distinct digits is:

$$5! = 5 \times 4 \times 3 \times 2 \times 1 = 120$$

So, there are 120 such numbers, and they are ordered from smallest to largest.

**Finding the 50th Permutation**

To find the 50th permutation, I think it would be helpful to understand how the permutations are ordered.

Each unique starting digit (first digit) can lead to $4! = 24$ permutations because there are 24 ways to arrange the remaining four digits.

Since $24 \times 2 = 48$, the 50th permutation falls into the range where the first digit is 5 (because the first 48 permutations must have 1, 2, or 3 as the first digit).
Wait, let's check:

- The first 24 numbers start with 1.

- The next 24 numbers start with 2.

- The next 24 numbers start with 3.

- The next 24 numbers start with 4.

- The next 24 numbers start with 5.

So:

- Numbers 1 to 24: first digit = 1

- Numbers 25 to 48: first digit = 2

- Numbers 49 to 72: first digit = 3

Aha, so the 50th permutation is between numbers 49 and 72, which corresponds to the first digit being 3.

**Narrowing Down Further**

Now, I need to find the 50th permutation. Since the first 48 are covered (24 starting with 1 + 24 starting with 2), the 49th to 72nd permutations start with 3.
Thus, the 50th permutation is the **2nd permutation** in the set of permutations starting with 3 (since $48 + 2 = 50$).
To find permutations starting with 3, the remaining digits are 1, 2, 4, 5. Now, the task reduces to finding the 2nd smallest permutation of the digits 1, 2, 4, 5 with 3 fixed as the first digit.

**Finding the 2nd Smallest Permutation of 1, 2, 4, 5**

First, find all permutations of 1, 2, 4, 5 ordered from smallest to largest. To find the 2nd permutation, let's list them step by step.
The smallest permutation (1st) will have the smallest possible digits in the subsequent places.

1. **1st permutation:** Start with 1, remaining digits 2, 4, 5 in ascending order: **1 2 4 5**.

2. **2nd permutation:** Keep 1 as the second digit, swap the last two: **1 2 5 4**.

Thus, the second permutation starting with 3 (which is the 50th overall) is **3 1 2 5 4**.

**Final Answer**

After this reasoning, the 50th five-digit integer using each of the digits 1, 2, 3, 4, and 5 exactly once, in ascending order, is

$$\boxed{31254}$$

*Figure 13.* A rollout elected by CBS, where the answers and CoT are both right.

