# OpenReview forum: "Contextual Rollout Bandits for Reinforcement Learning with Verifiable Rewards"
_ICML.cc/2026/Conference — ICML 2026 regular_

### Official Review · Reviewer_9N2H · 2026-03-04

**Soundness:** 3
**Presentation:** 3
**Significance:** 2
**Originality:** 3
**Overall Recommendation:** 5
**Confidence:** 3

**Summary:**

This paper approaches two problems in most existing RLVE methods, including indiscriminate rollout usage within a group and the issue of the restricted data horizon. Specifically, many RLVE methods rely on group-relative policy optimization (GRPO), which may incorporate all generated samples for the following training. However, some of those samples are low-quality, and thus training on them may incur unnecessary computational cost and degrade the policy training. The other issue is that RLVR approaches rely on the latest batch of rollouts, but some historical rollouts could also be informative. To address these two issues, this paper uses a contextual bandit framework to model the rollout scheduling and propose a neural scheduler. The authors show how to plug the proposed method into the RLVR pipeline. Both theoretical motivation and empirical experiments are provided.

**Compliance With Llm Reviewing Policy:**

Affirmed.

**Final Justification:**

My questions have been fully resolved in the authors' rebuttal. I believe this is a solid paper and will increase my score to 5.

**Key Questions For Authors:**

- Please refer to my question in weaknesses on how large $\tilde{d}$  can be.

- Can authors explain more on the setup of eq(8) and eq(9)?

**Limitations:**

I don't see a section (e.g., Conclusion section) that authors explicitly discuss the limitations.

**Strengths And Weaknesses:**

**Strengths:**
 - The paper formulates the rollout scheduling in RLVR in the contextual bandit framework and proposes a neural scheduling approach plugged into the RVLR methods. The proposed approach addresses intra-group scheduling and global scheduling problem issues in RLVR methods.

- The authors provide theoretical guarantees for both local and global cases in a simplified setting.

- The authors conduct extensive experiments on 6 mathematical reasoning benchmarks, which demonstrate the effectiveness of the proposed method across many RLVR optimization algorithms.

- The paper is generally well-written and well-organized.


**Weaknesses:**
- I am concerned about the significance of Theorem 5.3. In particular, Theorem 5.3 suggests that a larger buffer size ensures a better *optimal* performance of the policy model. However, I feel this is pretty straightforward, especially considering that the theorem is proved for the optimal performance. In other words, it is not surprising that an oracle (which can act optimally) can do better when the oracle is given more options to choose from (i.e., a larger buffer size).

- This paper only provides theoretical guarantees for a simpler setting where only one rollout is selected per round, while the algorithm is designed for the general case.

- This paper lacks a clear discussion on how large $\tilde{d}$ can be. Since $\tilde{d}$ depends on matrix $H$, but matrix $H$ is $TM \times TM$. In other words, $\tilde{d}$ depends on horizon $T$. Then, I am not sure whether $O(\sqrt{T})$ regret bound is proper. In (contextual) linear bandits setting, $H$ is a $d \times d$ matrix, and the final regret bound is linear in $d$. If so, does the regret bound in this bound may also scale linearly in $TM$?

---

> ### Author Rebuttal · Authors · 2026-03-31
>
> Thank you for your time and detailed feedback. In this rebuttal, we further discuss the growth of the effective dimension $\tilde d$, the design of Eqs. (8) and (9), and the scope of the current theory. We hope these address your concerns and look forward to your feedback.
>
> > Q1 & W3: Please refer to my question in weaknesses on how large $\tilde{d}$ can be.
>
> **The effective dimension $\tilde d$ is data-dependent and may grow only mildly with $T$ in favorable regimes.** We agree that $\tilde d$ depends on the realized data, but this does not mean it must grow linearly with $T$. In favorable structured regimes, it can remain controlled. For example, [1] (Remark 4.4) and [2] (Remark 3.3) show that when the NTK matrix of arm representations has an underlying low-dimensional structure, $\tilde{d}$ can remain bounded. Therefore, we follow prior neural contextual bandit analyses [1,2,3] and write the bound in terms of $\tilde d$ with the $\widetilde{O}(\sqrt{T})$ notation. We will clarify this data-dependent behavior and these favorable regimes in the revised version.
>
> References:
> [1] Zhou, Dongruo, et al. Neural Contextual Bandits with UCB-based Exploration. ICML, 2020.
> [2] Zhang, Weitong, et al. Neural Thompson Sampling. ICLR, 2021.
> [3] Ban, Yikun, et al. EE-Net: Exploitation-Exploration Neural Networks in Contextual Bandits. ICLR, 2022.
>
>  > Q2: Can authors explain more on the setup of eq(8) and eq(9)?
>
> **Eq. (8) explains how $\epsilon$-greedy selection is performed within each round**: the scheduler sequentially selects multiple rollouts from the context set without replacement. At each selection, with probability $\epsilon_t$ it explores by choosing one rollout from the newest data (i.e., the rollout with minimum $f_d(x)$), and with probability $1-\epsilon_t$ it exploits by choosing the rollout with the highest scheduler score. We use the newest rollouts for exploration because they have not been seen before and provide the strongest exploration signal.
>
> **Eq. (9) explains how the exploration probability changes over time.** We introduce a warmup threshold $T_{\mathrm{warm}}$. When $t < T_{\mathrm{warm}}$, the scheduler is still poorly learned, so we set the exploration probability to 1 and use pure exploration. When $t > T_{\mathrm{warm}}$, the exploration probability is linearly decayed until it reaches $\epsilon_{\min}$, so training gradually shifts to scheduler-guided exploitation.
>
> > W1: I am concerned about the significance of Theorem 5.3. In particular, Theorem 5.3 suggests that a larger buffer size ensures a better optimal performance of the policy model. However, I feel this is pretty straightforward, especially considering that the theorem is proved for the optimal performance. In other words, it is not surprising that an oracle (which can act optimally) can do better when the oracle is given more options to choose from (i.e., a larger buffer size).
>
> **The significance of Theorem 5.3 is that it shows, under a realizable neural scheduler with finite width and depth, enlarging the buffer can improve the achievable upper bound on performance.** We agree that if the scheduler class were an unrestricted oracle that could assign arbitrary scores to arbitrary rollouts, then Theorem 5.3 would be trivial. However, this monotonicity does not hold for an arbitrary restricted scheduler class. To illustrate this, consider a linear scheduler $s_w(x)=w^\top x$. Suppose the current round produces two fresh rollouts $x_1=(1,0)$ and $x_2=(-1,0)$ with reward $1$, while the previous round contributes two stale rollouts $x_3=(2,0)$ and $x_4=(-2,0)$ with reward $0$. In the small-buffer case with buffer size $2$, the scheduler only sees $\{x_1,x_2\}$ and achieves reward $1$. In the larger-buffer case with buffer size $4$, it sees $\{x_1,x_2,x_3,x_4\}$: any $w>0$ selects $x_3$, any $w<0$ selects $x_4$, and $w=0$ gives a tie with expected reward $1/2$. Hence, within this restricted class, enlarging the buffer lowers the optimal value.
>
> > W2: This paper only provides theoretical guarantees for a simpler setting where only one rollout is selected per round, while the algorithm is designed for the general case.
>
> **In the revised version, we will extend Appendix C to provide a more formal analysis of the batch-selection setting.** Our current theorem focuses on the single-rollout case ($K=1$). For the more general group-selection setting, under the Appendix C assumptions that the subset-level reward is additive over selected samples and that individual sample rewards are observed after selecting a group, the problem can be viewed as a contextual semi-bandit. The group regret can then be decomposed into the sum of regrets over the selected samples, which retains the same sublinear $\tilde O(\sqrt{T})$ dependence on the horizon up to an additional dependence on the group size $K$. Our appendix states a conservative factor-$K$ extension of the single-rollout bound, and we will formalize this extension in the revised version.

---

> > ### Author Rebuttal · Reviewer_9N2H · 2026-04-02
> >
> > My questions have been fully resolved in the authors' rebuttal. I believe this is a solid paper and will increase my score to 5.

---

> > > ### Author Response · Authors · 2026-04-03
> > >
> > > Thank you for your timely and positive reply. We are very grateful to hear that our rebuttal has fully addressed your concerns and that your overall assessment has increased to 5. We greatly appreciate the time, effort, and thoughtful comments you devoted to our submission, which have been very helpful in improving the paper.

---

### Official Review · Reviewer_Fhc9 · 2026-03-06

**Soundness:** 2
**Presentation:** 2
**Significance:** 3
**Originality:** 3
**Overall Recommendation:** 5
**Confidence:** 2

**Summary:**

In this paper, the authors address a limitation of reinforcement learning with verifiable rewards (RLVR), a training paradigm commonly used to improve the reasoning capabilities of large language models. In existing approaches, rollouts are typically used regardless of their quality and are not stored for later reuse. To address these shortcomings, the authors formulate the training process as a contextual bandit problem, where each rollout corresponds to an arm of the bandit and the reward reflects the resulting performance gain. The authors further derive a sublinear regret bound for their method and demonstrate superior training performance and efficiency compared to existing approaches.

**Compliance With Llm Reviewing Policy:**

Affirmed.

**Final Justification:**

The paper introduces an interesting idea supported by empirical validation. My concerns have been fully addressed and adding them to the main text will make this a valuable contribution.

**Key Questions For Authors:**

1. Can the authors clarify the gap between the reward used in theory (Vt) and in practice (Equation 5), and discuss whether the regret bounds still hold approximately under the practical reward definition?
2. Is there an ablation on the impact of the epsilon-greedy exploration strategy, given that it is omitted in the theoretical analysis but present in the implementation?

**Limitations:**

As far as I can tell, the paper does not include a discussion of the limitations of the proposed method. This discussion could address the connection between the setting used in the theoretical section and the one used in practice.

**Strengths And Weaknesses:**

The authors’ idea of formulating the choice of training data selection as a contextual bandit problem is well-motivated and elegant. Treating each rollout as an arm whose reward reflects the induced policy performance gain provides a principled and unified framework that naturally accommodates both intra-group noise filtering and global historical reuse within a single decision-making paradigm. Providing a sublinear regret adds a theoretical flavor to the presentation of their methods. The empirical results are strong and support the authors’ main claims: performance gains and computational efficiency. Additionally, the ablation study provides useful insights into the contribution of different components of the method. However, the theoretical analysis simplifies away key aspects of the full method — notably, the reward used in theory (Vt) differs from the practical reward in Equation (5), and the epsilon-greedy exploration is omitted entirely. It is unclear how tightly the guarantees reflect actual behavior. On presentation, the introduction is dense for readers less familiar with RLVR, as a lot of topic-specific notation is defined without an illustrative example, e.g., the tripled-indexed x's and o's in Section 3.1.2. make it difficult to grasp the pipeline's flow. At last, I would have liked to see a brief discussion of the limitations of the proposed method

---

> ### Author Rebuttal · Authors · 2026-03-31
>
> Thank you for your time and detailed feedback. In this rebuttal, we further discuss the practical reward, the role of $\epsilon$-greedy exploration, the presentation of the rollout pipeline, and the current limitations of the method. We hope these address your concerns and look forward to your feedback.
>
> > Q1& W1: Can the authors clarify the gap between the reward used in theory (Vt) and in practice (Equation 5), and discuss whether the regret bounds still hold approximately under the practical reward definition?
>
> **Using Eq. (5) does not change the conclusion of Theorem 5.4: the regret bound retains the same $\tilde O(\sqrt{T})$ order, up to an additional constant factor**. Theorem 5.4 does not rely on a specific reward form; it only requires the bounded-reward condition in Assumption 5.1. Therefore, by applying a simple transformation to the reward of Eq. (5), the corresponding cumulative regret can be reduced to the setting handled by Theorem 5.4. Specifically, let $\lambda_t = w_e I[E_t > e_{\min}]$. The conditional expected reward induced by Eq. (5) is
> $g_t(x) = \mathbb{E}[V_{t+1} - V_t - \lambda_t (E_{t+1} - E_t) \mid \mathcal{A}(x,\theta_{t-1})].$
> Here $\mathcal{A}(x,\theta_{t-1})$ denotes the policy parameter obtained by updating $\theta_{t-1}$ with rollout $x$. By Lemma B.2, $g_t(x)$ is bounded in $[-(1+w_e \log N), 1+w_e \log N]$. Let $B = 1 + w_e \log N$ and define $\bar g_t(x) = \frac{g_t(x)+B}{2B}$. Then $\bar g_t(x) \in [0,1]$ and the cumulative regret of $\bar g_t$ is $\tilde{O}(\sqrt{T})$. Since the regret of $g_t$ differs from that of $\bar g_t$ only by the constant factor $2B$, Eq. (5) yields the same regret order as Theorem 5.4, with only an additional constant factor $2B$.
>
> > Q2 & W2: Is there an ablation on the impact of the epsilon-greedy exploration strategy, given that it is omitted in the theoretical analysis but present in the implementation?
>
> **A properly decayed $\epsilon$-greedy schedule does not change the order of the regret bound.** Let $a_{t}^{o}$, $a_{t}$, $a_{t}^{g}$, and $a_{t}^{e}$ denote the optimal arm, the selected arm, the greedy arm, and the exploration arm, respectively, and let $I_{t}$ be a Bernoulli variable indicating whether exploration is taken at round $t$. Then the regret at round $t$ satisfies $r(a_{t}^{o}) - r(a_{t}) = (r(a_{t}^{o}) - r(a_{t}^{g})) + I_{t}(r(a_{t}^{g}) - r(a_{t}^{e}))$. Here, the first term is the regret without $\epsilon$-greedy, and the second term is the additional penalty introduced by $\epsilon$-greedy. Since $r(a_{t}^{g}) - r(a_{t}^{e}) \leq 1$, the extra exploration regret over $T$ rounds is bounded by $\sum_{t=1}^{T} I_{t}$. If we set $\epsilon_{t} = 1/t$, Bernstein's inequality gives $\sum_{t=1}^{T} I_{t} = O(\log T)$ with probability at least $1-\delta$. Hence the overall regret still retains the same $\tilde O(\sqrt{T})$ order as in Theorem 5.4. Empirically, we compare CBS with and without $\epsilon$-greedy under the Qwen3-4B setting while keeping the other training settings unchanged. Removing $\epsilon$-greedy decreases the average score from 42.41 to 42.27, which is consistent with its practical benefit.
>
> **In the revised version, we will provide a single-rollout analysis fully aligned with the empirical setting, incorporating both Eq. (5) and $\epsilon$-greedy.**
>
>
> > W3: On presentation, the introduction is dense for readers less familiar with RLVR, as a lot of topic-specific notation is defined without an illustrative example, e.g., the tripled-indexed x's and o's in Section 3.1.2. make it difficult to grasp the pipeline's flow.
>
> We apologize for the confusion. **In the revised version, we will add a clearer visual illustration of rollout data; the current figure can be found [here](https://anonymous.4open.science/r/ICML-2026-Rebuttal-7779/).**
> Specifically, for $x_{i,j}^t$, the three indices denote the training round, the prompt, and the generated response. The superscript $t$ denotes the training round, the subscript $i$ indexes a prompt in the batch, and the subscript $j$ indexes one of the generated responses for that prompt.
>
>
> > W4: At last, I would have liked to see a brief discussion of the limitations of the proposed method
>
> **A main limitation is the expressiveness of the current 10-dimensional arm representation.** Specifically, each rollout is represented by a lightweight 10-dimensional feature vector. This design is chosen for computational efficiency, but it may not fully capture deeper signals related to data value, such as richer semantic cues or the model's internal confidence. This is the current trade-off between efficiency and representation expressiveness. Future work can incorporate richer semantic representations from the LLM policy itself for more fine-grained rollout selection. We will add this limitation discussion more explicitly in the revised version.

---

> > ### Author Rebuttal · Reviewer_Fhc9 · 2026-04-02
> >
> > Thank you for the clarification. My concerns are fully resolved. I update my score to a 5.

---

> > > ### Author Response · Authors · 2026-04-02
> > >
> > > Thank you for your timely and positive reply. We are very grateful to hear that our rebuttal has addressed your concerns and that your overall assessment has increased to 5. At the moment, the score shown on OpenReview may not yet reflect this update. If convenient, could you please edit your previous review to reflect this change? Thank you again for your time, effort, and thoughtful comments, which have been very helpful in improving the paper.

---

### Official Review · Reviewer_jpwc · 2026-03-07

**Soundness:** 3
**Presentation:** 3
**Significance:** 3
**Originality:** 3
**Overall Recommendation:** 5
**Confidence:** 1

**Summary:**

This paper proposes to use a contextual bandit-based scheduling technique to select rollouts from the buffer. Specifically, they developed a loss function and a ten-dimensional embedding of the rollouts. Then they propose a method to select the rollouts so that they can maximize the corresponding performance.

In addition to the methodology, they have shown the following things:
1. Theoretically, they characterize this problem as a contextual bandit problem, and they provide a guarantee that this algorithm can achieve sublinear regret.
2. Empirically, they implement their algorithms into mathematical reasoning datasets, and compared to traditional approaches, their methods have good performance.

**Compliance With Llm Reviewing Policy:**

Affirmed.

**Final Justification:**

My concerns of this paper were addressed. And I have updated my score

**Key Questions For Authors:**

I have identified the following questions regarding this paper:

1. I realize that the settings in the theoretical results and in the numerical experiments are not the same. Is it possible to unify the setting between the theoretical results and the empirical results?
2. From your numerical experiments, I identify that the improvement of algorithms with CBS is not that significant compared to the same algorithms without CBS. Is it possible that these improvements are not because the CBS provides a better rollout set, but because of something else?

**Limitations:**

yes

**Strengths And Weaknesses:**

Strength:

1. This paper is well written.
2. The introduction of the problem setting is clear, and the results and assumptions are stated in a clear way.
3. The numerical experiments are promising and demonstrate that the algorithms work well in practice.

Weakness:

1. The theoretical results adopt the Neural Tangent Kernel regime. However, this cannot fully characterize the learning procedure with neural networks.

---

> ### Author Rebuttal · Authors · 2026-03-31
>
> Thank you for your time and detailed feedback. In this rebuttal, we further discuss the gap between the theoretical and empirical settings, the significance of the observed gains, and the scope and limitations of the NTK-based analysis. We hope these address your concerns and look forward to your feedback.
>
> > Q1: I realize that the settings in the theoretical results and in the numerical experiments are not the same. Is it possible to unify the setting between the theoretical results and the empirical results?
>
> **The current mismatch does not change the main theoretical conclusion, and we will make this connection more explicit in the revised version through an analysis that is more fully aligned with the empirical setting.** Specifically, for the current single-rollout analysis, mismatches come from omitting $\epsilon$-greedy and using a simplified reward function. In our responses to Q1 \& W1 and Q2 \& W2 of Reviewer Fhc9, we clarify that these mismatches do not affect the order of the regret bound. For the general multi-rollout selection setting, obtaining a conclusion analogous to Theorem 5.4 requires assumptions on the group reward. As discussed in our response to W2 of Reviewer 9N2H, under the Appendix C assumptions that the subset-level reward is additive over selected samples and that individual sample rewards are observed after selecting a group, the problem can be viewed as a contextual semi-bandit, which introduces at most an additional dependence on the group size $K$ without changing the regret order. We will incorporate these points into a more formal batch-selection discussion in the revised version.
>
>
> > Q2: From your numerical experiments, I identify that the improvement of algorithms with CBS is not that significant compared to the same algorithms without CBS. Is it possible that these improvements are not because the CBS provides a better rollout set, but because of something else?
>
> We would like to clarify that the following evidence points to the scheduler itself as the source of the observed gains.
> - **Nontrivial overall gains.** Table 2 shows that CBS and CBS\* yield nontrivial overall improvements over their paired RLVR baselines. Averaged across settings, CBS improves GRPO, DAPO, and GSPO by 4.0%, 8.5%, and 1.7%, respectively, while CBS\* improves them by 5.0%, 5.8%, and 2.0%.
> - **Ablation evidence.** As shown in Figure 3, replacing the learned neural scheduler with random selection leads to clear drops (GRPO: 42.41→41.41, DAPO: 43.93→39.67, GSPO: 43.92→40.55), which suggests that the improvements are closely related to better rollout selection rather than unrelated factors.
> - **Efficiency benefit.** CBS\* only keeps the top-$p$% rollouts from the latest batch for policy optimization, so its benefit should not be viewed only from the performance perspective: it achieves non-trivial improvement while also reducing the average policy-optimization time by 50.4%, according to Figure 2.
> - **PO dependence.** Some variation in gain magnitude across methods appears to be natural, since CBS and CBS\* are data-selection plugins and their realized benefit depends on how rollout utility interacts with the underlying policy optimization objective.
>
>
> > W1: The theoretical results adopt the Neural Tangent Kernel regime. However, this cannot fully characterize the learning procedure with neural networks.
>
> **We would like to clarify that our goal is not to claim that NTK fully captures the RLVR training pipeline, but to show that under a standard and well-accepted approximation, the proposed scheduler admits a principled regret guarantee.** We agree that the NTK regime is only an approximation and cannot fully characterize the learning dynamics of finite-width MLPs. In general, the training dynamics of finite-width neural networks are still not well understood theoretically, and a full characterization remains an open problem. As a result, some degree of simplification is typically unavoidable when deriving regret guarantees. In this paper, we adopt the NTK framework as a tractable theoretical tool rather than an exact description of practical training. Notably, this type of NTK-based analysis is also widely used in the contextual bandit literature for studying nonlinear function approximation [1,2,3].
>
> References:
> [1] Zhou, Dongruo, et al. Neural Contextual Bandits with UCB-based Exploration. ICML, 2020.
> [2] Zhang, Weitong, et al. Neural Thompson Sampling. ICLR, 2021.
> [3] Ban, Yikun, et al. EE-Net: Exploitation-Exploration Neural Networks in Contextual Bandits. ICLR, 2022.

---

> > ### Author Rebuttal · Reviewer_jpwc · 2026-04-03
> >
> > Thanks for the detailed response. My concerns have been addressed. And I have updated my score.

---

> > > ### Author Response · Authors · 2026-04-04
> > >
> > > Thank you for your timely and positive reply. We are very grateful to hear that our rebuttal has addressed your concerns and that your overall assessment has increased. We greatly appreciate the time, effort, and thoughtful comments you devoted to our submission, which have been very helpful in improving the paper.

---

### Official Review · Reviewer_3xpW · 2026-03-07

**Soundness:** 2
**Presentation:** 3
**Significance:** 2
**Originality:** 2
**Overall Recommendation:** 4
**Confidence:** 4

**Summary:**

This paper improves RLVR for large language models by addressing inefficient and noisy use of rollouts during training. The authors formulate rollout scheduling as a contextual bandit problem and propose a neural framework that adaptively selects high-value rollouts and reuses useful historical ones. With theoretical guarantees and experiments on six math reasoning benchmarks, the method shows better performance and training efficiency than existing RLVR approaches.

**Compliance With Llm Reviewing Policy:**

Affirmed.

**Final Justification:**

I thank the authors for the detailed response and additional experimental results. My concerns have been addressed, and I have updated my score accordingly.

**Key Questions For Authors:**

The paper presents a comprehensive study including problem formulation, method design, theoretical analysis, and extensive experiments. However, my main concerns relate to the correctness and justification of the proposed method.

1. Since the reward of the same arm may change across multiple samplings and may become incremental as it is reused, the rewards from the same arm may not be independent. Does this violate the standard assumptions in contextual bandits, and how does it affect the theoretical guarantees?

2. The method reuses historical rollouts, while algorithms such as GRPO are on-policy. This may introduce high variance due to importance sampling ratios. How is this issue addressed, and is the reuse of old rollouts theoretically or empirically justified?

**Limitations:**

YES

**Strengths And Weaknesses:**

**Strengths**
1. The paper proposes to formulate rollout scheduling in RLVR as a contextual bandit problem and introduces a neural scheduler to select high-value rollouts. This perspective is novel and provides an interesting direction for improving sample efficiency in RLVR training.

2. The paper conducts extensive experiments, evaluating the proposed method with multiple RL optimization methods and across several mathematical reasoning benchmarks. The results show improvements in both performance and training efficiency, providing solid empirical support for the proposed approach.


**Weaknesses**
1. Under the paper’s definition, the reward of an arm, i.e., a rollout, is defined as the performance gain between consecutive optimization steps. However, for the same arm, repeated sampling can lead to different reward values, and the performance gain may become incrementally smaller as the rollout is reused more times. This implies that rewards from the same arm are not independent across pulls, which appears inconsistent with the standard assumptions in contextual bandit formulations.

2. The method reuses historical rollouts through a rollout buffer. However, algorithms such as GRPO are essentially on-policy. Reusing old rollouts introduces off-policy data, and the corresponding importance sampling ratios may lead to high variance during optimization. The paper does not seem to provide additional mechanisms to mitigate this issue. It is therefore unclear whether such reuse is theoretically or practically stable in this setting.

---

> ### Author Rebuttal · Authors · 2026-03-31
>
> Thank you for your time and detailed feedback. In this rebuttal, we further discuss the dynamic arm representation and the justification for reusing historical rollouts. We hope these address your concerns and look forward to your feedback.
>
> > W1 & Q1:  Since the reward of the same arm may change across multiple samplings and may become incremental as it is reused, the rewards from the same arm may not be independent. Does this violate the standard assumptions in contextual bandits, and how does it affect the theoretical guarantees?
>
> We thank the reviewer for this question. Our formulation does not violate contextual bandit assumptions, **because the arm representation evolves dynamically, so a reused rollout will not be presented as the same arm twice.**
>
> **Motivation of dynamic arm representation.** In the most faithful formulation, the arm representation should also include the current policy parameters, because the reward depends on both the rollout and current policy state. Reward changes across rounds are naturally captured by context changes. However, using the full model parameters is computationally infeasible. We therefore approximate this ideal context with lightweight dynamic features that include a rollout-specific projection of the policy state and data-usage history.
>
> **Detailed update mechanism.** Among the 10 features in Table 1, four update across rounds. `Entropy` and `Clip ratio` can be viewed as lightweight surrogates of the current model state projected onto that rollout, and are recomputed from the *current* policy whenever the rollout is used (see footnote 2 of Table 1). In principle, these two metrics could be refreshed at every step, but this would incur substantially higher computational cost, so we update them only when the rollout is revisited. `Usage count` records how many times the rollout has been selected and increases by 1 after each use, while `Sample age` records how many rounds have elapsed since the rollout is generated and increases by 1 at each round.
>
> **Consistency with contextual bandits.** In the contextual bandit framework, arm sets and contexts may vary arbitrarily across rounds. Different rewards for a reused rollout therefore arise because the scheduler faces different contexts, not because the same arm produces different rewards. Since Theorem 5.4 allows adversarial context sequences, this dynamic representation does not affect the guarantee under Assumptions 5.1 and 5.2.
>
> We will make this update mechanism more explicit in the revised version to avoid confusion.
>
> > W2 & Q2: The method reuses historical rollouts, while algorithms such as GRPO are on-policy. This may introduce high variance due to importance sampling ratios. How is this issue addressed, and is the reuse of old rollouts theoretically or empirically justified?
>
> The reuse of historical rollouts is justified by the following three observations.
>
> - **Existing group-relative policy optimization methods introduce the Clipping mechanisms that stabilize training on off-policy data** by masking gradients of tokens with extreme importance sampling ratios (`clip` of GRPO/DAPO/GSPO in Appendix A). Therefore, in standard implementations [1,2,3], including the one used in this paper, these policy optimization methods are not run in a strictly on-policy manner for efficiency. For example, the DAPO paper first collects rollouts for 512 prompts and then splits them into 16 batches of 32 prompts for 16 update steps [1]; after the first update step, the remaining batches are already no longer strictly on-policy.
>
>
> - **CBS further acts as a clip-aware scheduler.** As shown in Table 1 in our paper, the arm representation explicitly encodes policy perception metrics (`Entropy` and `Clip ratio`) — the latter directly reflects the fraction of tokens with extreme importance sampling ratios under the current policy. Since the scheduler is updated online based on observed performance gains, it continually re-evaluates historical rollouts and learns to downweight stale or high-variance samples rather than treating all buffered rollouts equally.
>
>
> - **Empirical verification under strict on-policy training**. To further verify stability, we compare GRPO and GRPO+CBS under a stricter on-policy setting (one gradient update per rollout phase). CBS yields consistent gains across all model sizes, suggesting that it remains effective and stable even when off-policy effects are minimized, and that its benefits come from adaptive rollout selection rather than exploitation of off-policy artifacts.
>
> |  | 1.7B | 4B | 8B |
> | --- | ---: | ---: | ---: |
> | GRPO | 26.54 | 40.07 | 42.44 |
> | GRPO + CBS | 27.68 | 41.19 | 45.24 |
> | Relative improvement | +4.30% | +2.80% | +6.60% |
>
> References:
> [1] Yu, Qiying, et al. DAPO: An open-source LLM reinforcement learning system at scale. NeurIPS, 2025.
> [2] Zheng, Chujie, et al. Group sequence policy optimization. arXiv, 2025.
> [3] Gao, Chang, et al. Soft adaptive policy optimization. arXiv, 2025.

---

> > ### Author Rebuttal · Reviewer_3xpW · 2026-04-02
> >
> > Thank you for the authors’ response. I partially accept the clarifications provided:
> >
> > 1. Regarding the off-policy issue, I acknowledge that there have been works introducing relaxation techniques to address practical concerns such as efficiency, and I find such engineering-oriented treatments acceptable. However, in this paper, this aspect is claimed as a key contribution, and I find the level of rigor insufficient. The clipping operation already exists in the original PPO algorithm; if clipping alone were sufficient to resolve distributional shift, it is unclear why PPO would not naturally extend to an off-policy setting. Moreover, when the distribution mismatch becomes large, clipping may simply lead to vanishing gradients. In that case, it is unclear what benefit is obtained from training on highly off-policy samples.
> >
> > 2. Regarding the bandit formulation, I partially accept the authors’ explanation, but I am not yet fully convinced.
> >
> > Overall, considering that the paper presents an interesting idea and is supported by fairly thorough experiments, I will increase my score to 3.

---

> > > ### Author Response · Authors · 2026-04-04
> > >
> > > Thank you very much for your timely reply and follow-up questions. In this response, we further clarify the role of CBS in handling historical rollouts with distribution shift and the consistency of our formulation with the contextual bandit framework.
> > >
> > > > Regarding the off-policy issue ...
> > >
> > > We appreciate this insightful comment and would like to clarify an important distinction: **CBS does not rely on clipping alone; it uses clipping signals as _input features_ for a learned neural scheduler that decides _whether to reuse_ each historical rollout.**
> > >
> > > The reviewer correctly notes that clipping alone is insufficient: when distribution mismatch is large, clipping leads to vanishing gradients and makes such samples unhelpful. This is precisely our motivation. Rather than training on all buffered rollouts with clipping as a passive safeguard, CBS actively **filters out** rollouts with large distribution shift before they enter training. The clip ratio is one of several dynamic features (alongside entropy, usage count, and sample age) fed to the scheduler, which learns from online policy feedback whether each rollout is worth reusing.
> > >
> > > To empirically verify that the scheduler (not clipping alone) drives the gains, we compare CBS with two ablated variants:
> > > - **`Random`**: Replaces the learned neural scheduler with a random sampler that reuses rollouts indiscriminately. The average relative improvement drops from **+5.3%** (Base+CBS) to **−1.7%** (Base+Random), confirming that naive reuse with only clipping as a safeguard is indeed harmful.
> > > - **`Static Arm`**: Removes the four dynamic features (entropy, clip ratio, usage count, sample age) while keeping the learned scheduler. Performance drops from **+5.3%** to **+3.6%**, showing that dynamic features capturing distribution shift are important for the scheduler to make informed decisions.
> > >
> > > Due to time constraints, we only report the 1.7B and 4B results below. `Method` denotes the PO method and model size, `Base` the standard PO method, and each entry reports `average test score (relative improvement over the corresponding Base)`.
> > >
> > > |Method|Base|Base+Random|Base+Static Arm|Base+CBS|
> > > |---|---|---|---|---|
> > > |GRPO (1.7B)|28.91 (+0.0%)|28.24 (-2.3%)|28.77 (-0.5%)|30.17 (+4.4%)|
> > > |DAPO (1.7B)|25.92 (+0.0%)|26.86 (+3.6%)|29.30 (+13.0%)|30.00 (+15.7%)|
> > > |GSPO (1.7B)|30.87 (+0.0%)|31.02 (+0.5%)|31.62 (+2.4%)|31.82 (+3.1%)|
> > > |GRPO (4B)|41.23 (+0.0%)|41.41 (+0.4%)|41.82 (+1.4%)|42.41 (+2.9%)|
> > > |DAPO (4B)|41.33 (+0.0%)|39.67 (-4.0%)|44.50 (+7.7%)|43.93 (+6.3%)|
> > > |GSPO (4B)|44.11 (+0.0%)|40.55 (-8.1%)|42.90 (-2.7%)|43.92 (-0.4%)|
> > > |Avg. rel. improvement|+0.0%|-1.7%|+3.6%|+5.3%|
> > >
> > > These results address the reviewer's concern:
> > > 1. The `Random` baseline validates that **clipping alone does not justify rollout reuse** — random reuse with clipping hurts performance on average.
> > > 2. CBS's advantage comes from its **learned, feedback-driven scheduling policy** that avoids selecting highly off-policy rollouts, rather than from any assumption that clipping can handle arbitrary distribution shift.
> > >
> > > > Regarding the bandit formulation ...
> > >
> > > **Consistency with the contextual bandit formulation.** In a standard contextual bandit, at round $t$ the learner observes arms $\\{1,...,T\\}$ with context vectors $\\{h_{t,1},\dots,h_{t,K}\\}$, selects an arm $a_t$, receives noisy feedback from an unknown reward function $f(h_{t,a_t})$ [1,2], and minimizes cumulative regret $\sum_{t=1}^{T}(f(h_{t,a_t^o})-f(h_{t,a_t}))$, where $a_t^o$ is the optimal arms with highest reward. Under the single-rollout setting, our scheduler analogously observes rollouts $\\{x_{t,1},\dots,x_{t,K}\\}$ (i.e., arms $\\{1,..., T\\}$) with 10-dimensional representations $\\{h_{t,1},\dots,h_{t,K}\\}$, selects one rollout $x_t$, and receives noisy feedback from $f(h_t)=g(\mathcal A(\theta_{t-1}, x_t))$ (Eq. 12 of the paper), where optimal arm $x_t^o$ is the rollout that yields the largest resulting performance increase.
> > >
> > > **Context construction.** Ideally, the context for $x_{t,i}$ would be $h_{t,i}=(\theta_{t-1},x_{t,i})$, since the reward depends on both the current policy state and the rollout. Encoding $\theta_{t-1}$ is infeasible, so we use lightweight dynamic surrogate features. This is also standard in contextual bandits, which typically use observable features rather than a perfect but intractable arm representation [3,4].
> > >
> > > We sincerely thank you again for the time, effort, and thoughtful comments you devoted to our submission, and hope that these additional results can address your remaining concerns. We would be very grateful if you would consider revising your assessment in light of these new results.
> > >
> > > References:
> > > [1] Zhou et al. Neural Contextual Bandits with UCB-based Exploration. ICML 2020.
> > > [2] Zhang et al. Neural Thompson Sampling. ICLR 2021.
> > > [3] Li et al. A Contextual-Bandit Approach to Personalized News Article Recommendation. WWW 2010.
> > > [4] Wang et al. Learning Hidden Features for Contextual Bandits. CIKM 2016.

---

### Decision · Program_Chairs · 2026-04-30

**Decision:**

Accept (regular)

**Comment:**

This submission introduces a unified neural scheduling framework for Reinforcement Learning with Verifiable Rewards (RLVR), aiming to improve the reasoning capabilities of Large Language Models (LLMs). By formulating rollout scheduling as a contextual bandit problem, the authors propose a mechanism to adaptively select high-value rollouts, addressing the inefficiencies of existing methods that treat all rollouts uniformly and discard historical data.

At the beginning of the review process, several reviewers raised concerns regarding the theoretical analysis and an apparent gap between the theoretical guarantees and the experimental results. However, during the rebuttal phase, the authors provided comprehensive responses and additional clarifications that successfully addressed all of these points. The reviewers acknowledged that their concerns regarding the scaling of regret bounds and the neural scheduling architecture were fully resolved, leading to score increases toward Accept and Weak Accept. Overall,  the paper offers a well-motivated and technically sound solution to a relevant problem in RL and LLM alignment, and it is recommended for acceptance.

Recommendation: Accept.